# The UBP5 histone H2A deubiquitinase counteracts PRCs-mediated repression to regulate Arabidopsis development

James Godwin [1,7], Mohan Govindasamy [1,8], Kiruba Nedounsejian[1,8], Eduardo March [1], Ronan Halton[1], Clara Bourbousse[2], Léa Wolff[2], Antoine Fort[3], Michal Krzyszton [4], Jesús López Corrales[5], Szymon Swiezewski[4], Fredy Barneche[2], Daniel Schubert[6] & Sara Farrona [1] ✉

Polycomb Repressive Complexes (PRCs) control gene expression through the incorporation of H2Aub and H3K27me3. In recent years, there is increasing evidence of the complexity of PRCs' interaction networks and the interplay of these interactors with PRCs in epigenome reshaping, which is fundamental to understand gene regulatory mechanisms. Here, we identified UBIQUITIN SPECIFIC PROTEASE 5 (UBP5) as a chromatin player able to counteract the deposition of the two PRCs' epigenetic hallmarks in *Arabidopsis thaliana*. We demonstrated that UBP5 is a plant developmental regulator based on functional analyses of *ubp5*-CRISPR Cas9 mutant plants. UBP5 promotes H2A monoubiquitination erasure, leading to transcriptional de-repression. Furthermore, preferential association of UBP5 at PRC2 recruiting motifs and local H3K27me3 gaining in *ubp5* mutant plants suggest the existence of functional interplays between UBP5 and PRC2 in regulating epigenome dynamics. In summary, acting as an antagonist of the pivotal epigenetic repressive marks H2Aub and H3K27me3, UBP5 provides novel insights to disentangle the complex regulation of PRCs' activities.

Histones that form the nucleosome, i.e. basic units of the chromatin, are marked by an array of covalent marks, especially on histone amino terminal tails but also on globular domains. Histone marks impact chromatin structure, modify its packaging and act as an anchor for chromatin-related proteins, transcription factors and other components of the transcriptional machinery[1]. Therefore, different systems evolved in the eukaryotic nuclei to act as 'writers', able to deposit covalent chemical groups on specific histone residues, 'readers', which can directly bind and help to interpret histone marks, and 'erasers',

actively removing histone post-translational modifications. The orchestration of histone modifying enzymes allows for a highly dynamic chromatin regulation crucial to control nuclear structure and transcription[2]. Two important histone modifications that are well conserved between plants and animals are the trimethylation on the lysine 27 of the histone H3 (H3K27me3)[3] and the monoubiquitination of the histone H2A that in plants mostly occurs on the lysine 121 (H2Aub)[4].

H3K27me3 and H2Aub are deposited, both in plants and animals, by two major types of Polycomb repressive complexes (PRCs),

[1]School of Biological and Chemical Sciences, College of Science and Engineering, University of Galway, H91 TK33 Galway, Ireland. [2]Institut de Biologie de l'École Normale Supérieure (IBENS), École Normale Supérieure, CNRS, INSERM, Université PSL, Paris, France. [3]Dept. of Veterinary and Microbial Sciences, Technological University of The Shannon: Midlands, Athlone, Co., Roscommon, Ireland. [4]Laboratory of Seeds Molecular Biology, Institute of Biochemistry and Biophysics, PAS, Warsaw 02-106, Poland. [5]Molecular Parasitology Laboratory (MPL), Centre for One Health and Ryan Institute, School of Natural Sciences, University of Galway, Galway H91 DK59, Ireland. [6]Institute of Biology, Freie Universität Berlin, 14195 Berlin, Germany. [7]Present address: Donald Danforth Plant Science Center, St. Louis, MO 63132, USA. [8]These authors contributed equally: Mohan Govindasamy, Kiruba Nedounsejian. ✉e-mail: sara.farrona@universityofgalway.ie

respectively PRC2 and PRC1. PRC2 is a four-core subunit complex in which the catalytic component is a SET (Su(var), Enhancer of zeste, Trithorax) domain histone methyltransferase (HMT)[5,6]. Analyses in different plant genomes showed that PRC2 decorates approximately 20–25% of euchromatic genes with H3K27me3, which switches them off in response to internal and external cues[7,8]. In plants, PRC1 is formed by E3 ligases and other auxiliary proteins[5,9]. Both PRCs maintain an intricate relationship in which members of the two complexes can directly interact, have common associated proteins, and share target genes. This is also reflected in their activities as H3K27me3 can precede H2Aub (i.e. hierarchical model) or oppositely follows this modification on the chromatin. Furthermore, both marks can independently regulate different set of genes[7,9].

In animals, H2AK119ub can be erased by the Polycomb Repressive-Deubiquitinase (PR-DUB) complex[10]. This complex contains a DUB protein of the ubiquitin carboxy-terminal (UCH) family, which does not have an obvious orthologous in plants[11]. Indeed, the PR-DUB has not been described in plants so far, but two proteins of the UBIQUITIN PROTEASE (UBP) family, UBP12 and UBP13 redundantly mediate H2A deubiquitination[12,13] and interact with LIKE HETEROCHROMATIN PROTEIN 1 (LHP1)[12], a H3K27me3 reader and interactor of both PRC2 and PRC1 components[7,9]. UBP12/13 regulate a similar set of genes with PRC2 and PRC1[13].

To develop their activities, PRCs require a complex network of protein-protein interactions[7]. We and others recently demonstrated that PWWP-DOMAIN INTERACTOR OF POLYCOMBS1 (PWO1) is a key regulator of PRC2 activity, able to interact with the HMTs of the PRC2 complex[14] and to form part of the PEAT complex (PWO/PWWP-EPCRs (ENHANCER OF POLYCOMB RELATED)-ARIDs (AT-RICH INTERACTION DOMAIN-CONTAINING)-TRBs (TELOMERIC REPEAT BINDING)) involved in heterochromatin dynamics[15]. Still, we are far from understanding the molecular impact of the PWO1-PRC2 interaction.

Here we show that UBP5 regulates plant development and affects both H3K27me3 and H2Aub marks as well as the expression of a set of PRC2 target genes in *Arabidopsis thaliana* (Arabidopsis). Telobox and GAGA motifs, previously related to PRC2 recruitment[16,17], are among the most enriched signatures of UBP5 binding to the chromatin. The vast majority of UBP5 direct target genes showed either hyper-marking or de-novo marking by H2Aub in *ubp5* plants, altogether indicating that UBP5 acts as an eraser of this epigenetic mark. Thus, our data uncovers UBP5 as a new regulator of PRCs' activities, directly controlling H2Aub deubiquitination and affecting H3K27 trimethylation to regulate gene expression.

## Results

### UBP5 interacts with PRC2 and PWO1

We had identified the UBIQUITIN PROTEASE 5 (UBP5) protein as the most abundant interactor co-immunoprecipitated with Arabidopsis PWWP-DOMAIN INTERACTOR OF POLYCOMBS1 (PWO1)[18]. Furthermore, data mining of proteins in co-immunoprecipitation (co-IP) experiments with PEAT components also identified UBP5[15]. Therefore, we aimed to understand the link between UBP5, PWO1 and PRC2. Firstly, to elucidate the sub-cellular localisation of UBP5, transient inducible expression was performed using the β-estradiol–inducible *35S* promoter (*i35S*) fused to an *UBP5-GFP* (*i35S::UBP5-GFP*) construct in *Nicotiana benthamiana* (*N. benthamiana*) and found that UBP5 is exclusively nuclear, localises all over the nucleoplasm in a diffused way but not in the nucleolus (Fig. 1a). Further, we analysed the possibility of an interaction between UBP5 and PWO1 *in planta*. Using a similar approach, we co-expressed *PWO1-GFP* and *UBP5-mCherry* fusion proteins in *N. benthamiana*. It is noteworthy that, as previously shown for CLF[14,18], co-expression of both proteins modified UBP5 localisation recruiting it to PWO1-containing nuclear speckles. To a lower extent, co-localisation of both proteins was also observed all over the nucleoplasm (Fig. 1b)[14,18], but the formation of speckles was never

observed when *UBP5-GFP* was expressed alone (Fig. 1a). PWO1-UBP5 association in the speckles was demonstrated by Foster resonance energy transfer with acceptor photobleaching (FRET-APB). FRET-APB efficiencies for co-expressed samples within the speckles were significantly higher than the negative controls (PWO1-GFP and UBP5-GFP expressed without donor mCherry construct) (Fig. 1c), which may indicate a preferential association within the speckles and/or higher contact probability between the two proteins within the speckles. Yeast two-hybrid (Y2H) assays not only confirmed the interaction of UBP5 with PWO1 but also revealed its interaction with the PRC2 HMT subunit SWINGER (SWN) ΔSET (SWN clone lacking the SET domain[19]); (Fig. 1d). *In planta* interaction between SWNΔSET and UBP5 was further confirmed using co-IP assays in *N. benthamiana* (Fig. 1e). Therefore, our protein-protein interaction results suggest that UBP5 is an interactor of PWO1-PRC2 and thus may play a role in PRC-mediated regulation of gene expression. Furthermore, Y2H assays showed interaction of UBP5 with EMBRYONIC FLOWER 2 (EMF2), another PRC2 component[20], which further confirms the PRC2-UBP5 connection (Fig. 1d).

### UBP5 is an essential plant developmental regulator

To understand UBP5 molecular functions in Arabidopsis, we generated an *ubp5* deletion mutant line via the CRISPR/Cas9 system with two guide RNAs, which partially deleted both DUSP and UBP conserved domains (Supplementary Fig. 1a–c). The phenotypic analyses of *ubp5* mutant plants showed pleiotropic defects such as stunted growth due to the lack of apical dominance (Fig. 2a (i–iii)), shorter roots and hypocotyl length (Fig. 2a ii and 2b), floral architecture defects (Fig. 2a (v–vi)), fertilisation defects (Supplementary Fig. 1d) and poor pollen germination (Supplementary Fig. 1e). These results fit well with the fact that *UBP5* is expressed all over the plant (Supplementary Fig. 2a), suggesting that UBP5 acts as a developmental regulator at different stages of the plant life cycle. Stable transformation of *UBP5pro::gUBP5-eGFP* was able to fully rescue the developmental pleotropic phenotypes of *ubp5* (Fig. 2a (iv), 2b) and confirmed UBP5 nuclear localisation (Supplementary Fig. 2b). qRT-PCR analyses further showed no significant difference in the relative expression of *UBP5* between the wild-type background Col-0 and the complementation line *UBP5pro::gUBP5-eGFP;ubp5* (Supplementary Fig. 3a, b). Transcriptional analyses of *ubp5* seedlings showed that 345 genes were up-regulated, and 478 genes were down-regulated (Fig. 2c; Supplementary data 1). Misregulation of developmental genes including *SAMBA*[21], *URACIL PHOSPHORIBOSYLTRANSFERASE* (*UPP*)[22], *GAMETOPHYTE DEFECTIVE 1* (*GAF1*)[23], *GOLGI CANDIDATE 4* (*GC4*)[24] and *ACTIN 1* (*ACT1*)[25] may explain some of the observed *ubp5* mutant phenotypes (Supplementary Fig. 4a–e; Supplementary data 2). Gene Ontology (GO) analyses identified that genes associated with biotic and abiotic stress responses terms were significantly enriched among all *ubp5* mis-regulated genes (Fig. 2d). Consistently with previous studies showing that PRC2-associated components do not only regulate expression of genes related to plant development[13,26–28]. Therefore, our phenotypic and transcriptomic data highlight UBP5 key role in regulating Arabidopsis development and a possible dual role in regulating stress responses that will require further investigation.

### UBP5 deubiquitinates H2A

UBP5 was shown to be involved in de-ubiquitination of hexa-ubiquitin substrates both in vivo and in vitro[29] and other UBP family members have been linked to the histone monoubiquitination removal[12,30,31]. In addition, the existence of the interaction between UBP5, PRC2 components and PWO1 made us speculate that UBP5 may contribute to PRC-mediated histone monoubiquitination dynamics. Therefore, we analysed different histone marks abundance in *ubp5* and Col-0 seedlings by western blot (WB) assays and, in good agreement with UBP5 acting in H2Aub removal, we found that H2Aub bulk levels were more

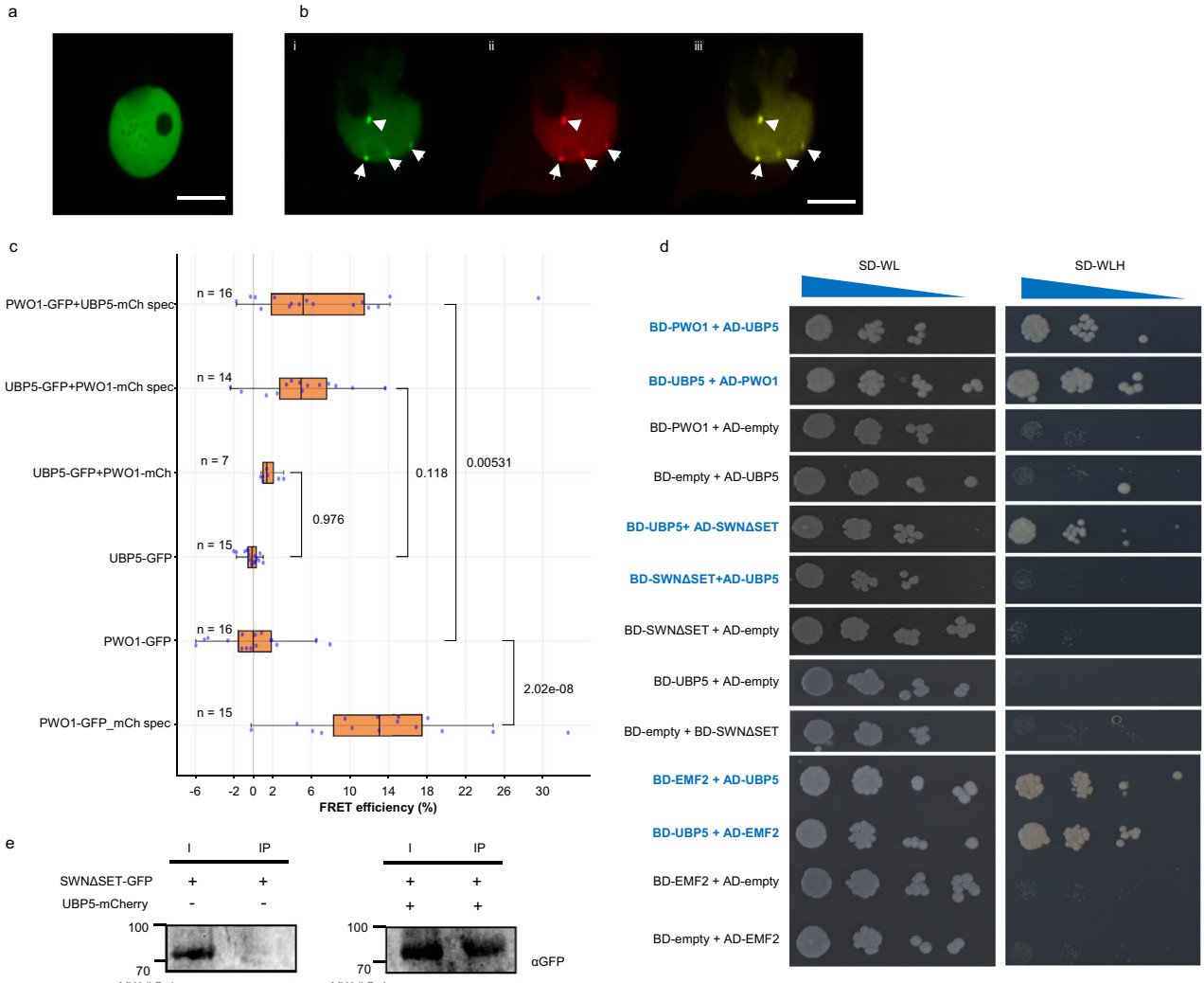

**Fig. 1 | UBP5 is a nuclear protein that interacts with PRC2 and colocalises with PWO1. a** Localisation of UBP5 in the nucleus of *N. benthamiana* epidermal cells. Data represent results of three independent experiments, scale bar indicates 10 μm. **b** Transient and inducible expression of *i35S::UBP5-GFP* and *i35S::PWO1-mCherry* in *N. benthamiana* epidermal cells by co-transformation (i, *i35S::UBP5-GFP;* ii, *i35S::PWO1-mCherry;* iii, overlay). Arrows indicate speckles. Data represent results of three independent experiments, scale bar indicates 10 μm. **c** FRET-APB measurements for nuclei exemplified in **b**, with a distinction for speckle (spec) and non-speckle localisation. *i35S::PWO1-GFP-mCherry* measurements in speckles were used as positive control. The respective average efficiency per nuclei (n) is given in the plot. For box plot, the middle line represents the median; the upper and lower lines are the first and third quartile (Q1 and Q3); the whiskers indicate the upper and lower limits of data spread by subtracting 1.5* interquartile range (IQR) from Q1 and adding 1.5* IQR to Q3.Statistical significance was calculated using one-way ANOVA and the *p*-values are indicated in the figure. Error bars correspond to SE. **d** Y2H analyses confirm UBP5-PWO1 interaction and show UBP5-SWN and UBP5-EMF2 interactions. Yeast cells containing the different construct combinations on selective medium for plasmids (-WL; -Tryptophan, Leucine) or for reporter gene activation (-WLH; -Tryptophan, Leucine, Histidine). Serial dilutions were used. BD, GAL4-DNA binding fusion; AD, GAL4-DNA activation domain fusion. SWNΔSET, SWN construct lacking the SET domain. **e** Co-IP analyses confirming SWN-UBP5 interaction. IP was performed with anti-mCherry antibody and proteins were detected by western blot with anti-GFP. I, 5% input; IP, immunoprecipitation. Two biological replicates were performed.

than 3-fold higher in *ubp5* (Fig. 3a). To gain insight into the affected loci, we profiled the genome-wide distribution of H2Aub in *ubp5* and Col-0 seedlings using ChIP-seq. Our H2Aub data in Col-0 seedlings showed a good overlap with previous published data (Supplementary Fig. 5) and, when compared to Col-0 seedlings, we observed a large increase in the number of genes marked by H2Aub in *ubp5* (21,017 in *ubp5* instead of 15,615 genes in Col-0; Supplementary data 3-4), which includes genes that differentially gained H2Aub in *ubp5* (n = 6201, called from now on de-novo marked genes; Fig. 3b), hence UBP5 is necessary to erase H2Aub in several thousands of genes. We then decided to conduct a comprehensive analysis on the H2Aub marked genes common to Col-0 and *ubp5* (n = 14816) and identified three different scenarios: genes that showed lower H2Aub levels in *ubp5* compared to Col-0 plants (n = 7150, hypo-marked genes), genes with higher H2Aub signal in *ubp5* compared to Col-0 (n = 3055, hyper-marked genes) and genes with non-significant differences between *ubp5* and Col-0 (n = 4611, unchanged genes) (Fig. 3c; Supplementary data 5).

To test whether UBP5 could act in H2Aub removal in *cis*, we further analysed the genome-wide association of UBP5-GFP in our *UBP5-pro:: gUBP5-eGFP;ubp5* line. Notably, UBP5 binding extends to a large part of the plant genome since the UBP5-GFP ChIP-seq profiling identified 8983 genes as direct UBP5 targets (Supplementary Fig. 6a−c; Supplementary data 6), which corresponds to ~27% of the total number of Arabidopsis genes according to TAIR 10 annotation[32]. More precisely, UBP5 directly targets 61% of hyper-marked genes, but only 9% of hypo-marked genes. In addition, among de-novo marked genes, 57% are also direct UBP5 targets. Therefore, a statistically significant number of UBP5 target loci gain the H2Aub mark in *ubp5* (1876 hyper-marked genes plus 3540 de-novo marked genes, 60% of UBP5 targets)

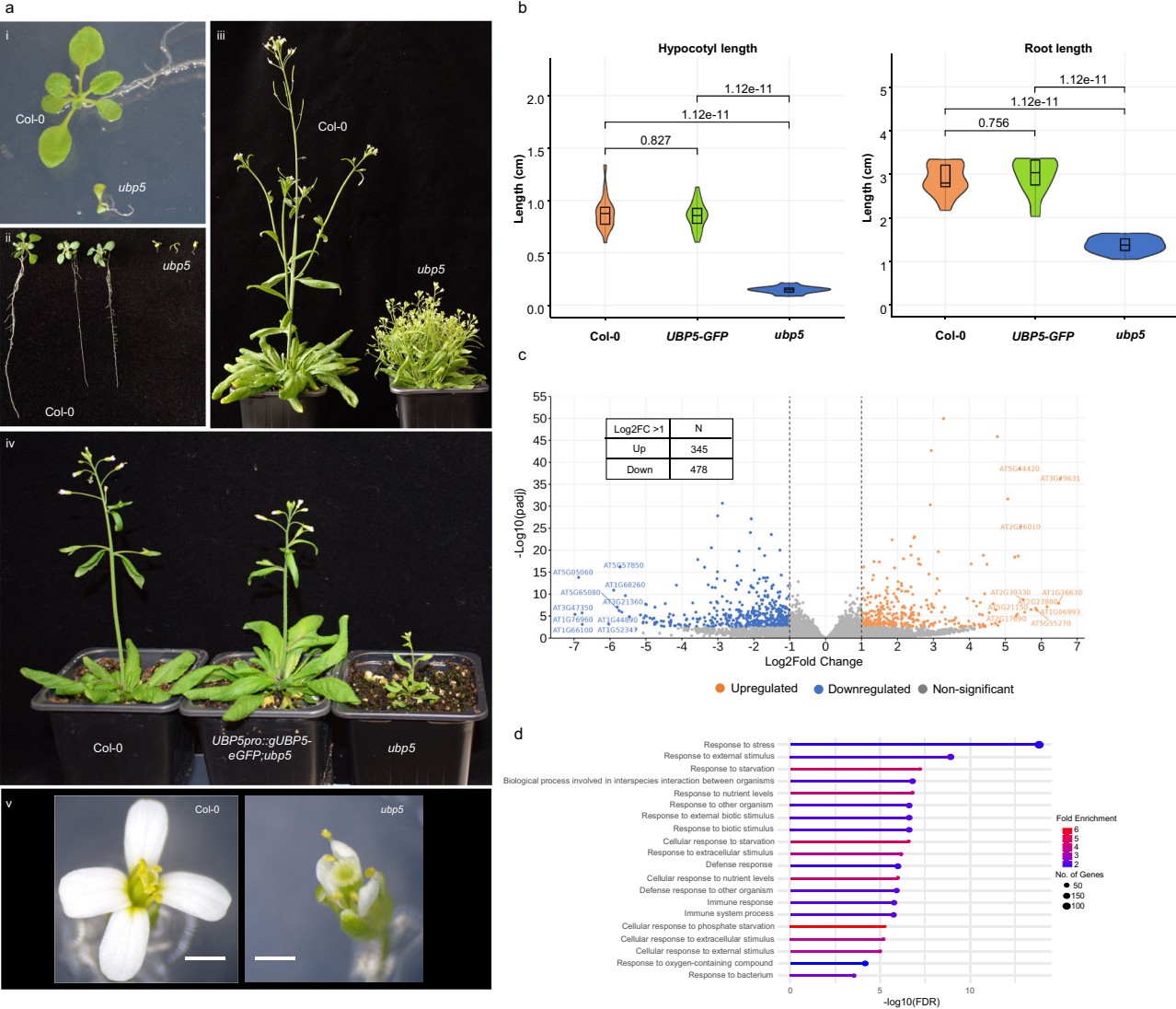

**Fig. 2 | UBP5 is an essential plant developmental regulator. a** Phenotypic characterisation of *ubp5* mutant line: i, smaller seedlings; ii, shorter primary roots; and iii) stunted and bushy growth (Note: in iii *ubp5* plant was 2 weeks older than the *Col-0* plant) compared to Col-0 plants; iv, complementation of *ubp5* mutant phenotypes in 4-week-old Arabidopsis plants (an *UBP5pro::gUBP5-eGFP* construct was used for the complementation of *ubp5*); v, floral phenotype of Col-0 and vi, *ubp5*. Scale bar = 2.5 mm. **b** Violin cum box plot showing hypocotyl and root length of Col-0, *UBP5-GFP* (complementation line) and *ubp5* measured 10 days after germination. The values were obtained from twenty independent seedlings (*n* = 20). The violin plots show the distribution pattern of data and are overlaid with boxplots. For box plots, the middle line represents the median; the upper and lower lines are the first and third quartile (Q1 and Q3); the whiskers indicate the upper and lower limits of

data spread by subtracting 1.5* interquartile range (IQR) from Q1 and adding 1.5* IQR to Q3. Statistical significance was calculated with one-way ANOVA and Tukey HSD and *p*-values are indicated. **c** Volcano plot of the misregulated genes between Col-0 vs *ubp5*. Genes with a p adjusted value (padj) lower than 0.05 are coloured. Up and down-regulated DEGs are shown in orange and blue colour dots respectively. The non-significant DEGs are represented as grey dots. The genes with Log2FC <1 or Log2FC < −1 (False Discovery Rate (FDR) < 0.05) were considered for further analysis and '*n*' denotes the number of genes. FDR values are derived from DESeq2 by adjusting *p*-values using Benjamini-Hochberg method. **d** Functional categorisation of *ubp5* misregulated (both upregulated and downregulated) genes based in ShinyGO v0.75 analysis. GO analysis of *ubp5* misregulated genes based on biological process (FDR < 0.05).

(Fig. 3d), pointing the key role of UBP5 in regulating H2Aub removal. Importantly, there is a sharp co-localisation between UBP5 chromatin association and domains where the H2Aub mark was gained in *ubp5* (Fig. 3e (i–ii); Supplementary Fig. 6a and 7a, b; Supplementary data 5). This frequent co-occurrence strongly argues in favour of a direct role for UBP5 in H2Aub deubiquitination at its binding sites (Fig. 3f). Further supporting this observation, there is an anticorrelation between the intensity of UBP5 binding and the presence of the H2Aub mark in Col-0 plants (Fig. 3f). In addition, an increase in H2Aub levels in *ubp5* is more evident at UBP5 target genes than for other, non-targets, H2Aub marked genes, in which even a slight significant decrease in the mark is observed (Fig. 3g and Supplementary Fig. 7c). Furthermore, overall increase of H2Aub levels was restored to Col-0 levels in the *UBP5pro::*

*gUBP5-eGFP;ubp5* line (Supplementary Fig. 7d) and inducing the expression of the *i35S::UBP5-GFP* construct in the nuclei of *N. benthamiana* leaves strongly decreased H2Aub levels (Fig. 3h). To confirm these observations, selected UBP5 targets that are H2Aub hypermarked in *ubp5* were further validated by ChIP–qPCR (Supplementary Fig. 7e). Overall, these results indicate that UBP5 acts in *cis* on the H2Aub mark by both maintaining H2Aub levels in a set of genes marked with this modification and erasing the H2Aub mark from a larger set of genes.

## UBP5 plays a role in transcriptional de-repression
Functional categorisation of UBP5 direct targets revealed that genes related to chromosome organisation, histone binding and chromatin

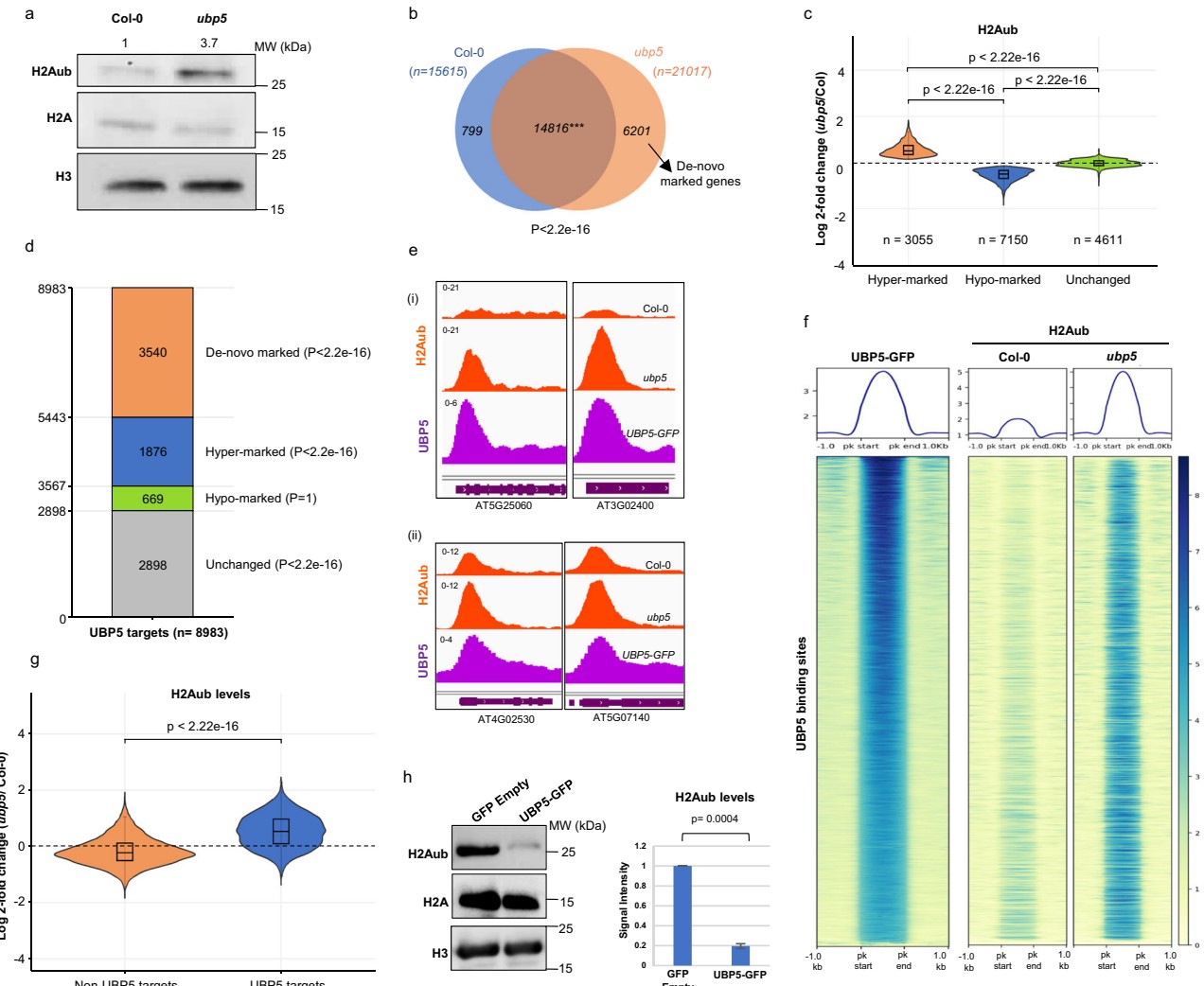

**Fig. 3 | UBP5 acts as a H2A deubiquitinase. a** Western blot analysis of H2Aub and H2A in Col-0 and *ubp5* seedlings. Histone H3 is used as a loading control. Numbers above blots denote relative H2Aub levels determined by ImageJ. Three independent experiments yielded consistent results. **b** Venn diagram illustrates overlap between H2Aub marked genes in Col-0 and *ubp5*, *n* = no. of genes. The genes are considered as marked when an overlapping H2Aub peak is present in at least two biological replicates based on MACS3 peak calling (*q* < 0.05 and score >30). Super exact test was conducted. **c** H2Aub levels in genes overlapping between Col-0 and *ubp5* (*n* = 14816) from Fig. 2B were categorised into hyper/hypo-marked and unchanged. Violin plots overlaid with boxplots show the distribution pattern of data. Displaying median (middle line), first and third quartiles (Q1 and Q3), with whiskers indicating data spread (1.5*IQR from Q1 and Q3). Statistical significance was tested using the Kruskal-Wallis test followed by Dunn's post hoc analysis. **d** Bar chart shows the number of H2Aub hyper-, hypo-, de-novo- and unchanged-H2Aub

marked genes represented within UBP5 target genes. Significance tested using Super Exact test. **e** IGV browser views of representative UBP5 target loci of (i) de-novo marked genes in *ubp5* and (ii) H2Aub hyper-marked genes in *ubp5*. **f** Heatmaps showing H2Aub distribution on genomic sequences targeted by UBP5, clustered based on higher to lower enrichment (top to bottom). **g** Average signal of H2Aub levels at non-UBP5 targets in Col-0 and *ubp5*. The violin cum box plots display median (middle line), first and third quartiles (Q1 and Q3), with whiskers indicating data spread (1.5*IQR from Q1 and Q3), two-sided Wilcoxon rank-sum test was performed. **h** H2A deubiquitination activity of UBP5. Transient expression of *i35S::UBP5-GFP* and *i35S::GFP* empty in *N. benthamiana* leaves. H2A deubiquitination was assessed by immunoblotting with α-H2Aub antibody; controls: α-H2A and α-H3 antibodies. Bar graph representing the relative H2Aub levels derived from band intensity. Error bars indicate SD between two biological replicates with two-tailed unpaired *t*-test, *p* values indicated above plot.

binding were significantly over-represented (Supplementary Fig. 8a–c). In addition to UBP5 direct protein-protein interaction with PWO1 and SWN chromatin factors, we found that UBP5 is also able to bind the chromatin of several loci encoding PRC2 subunits such as *CLF*, *EMF2*, *VERNALIZATION 2* (*VRN2*), *FERTILIZATION-INDEPENDENT ENDOSPERM* (*FIE*) and *MULTICOPY SUPPRESSOR OF IRA* 1 (*MSI1*) as well as PRC1 subunit encoding gene *B LYMPHOMA Mo-MLV INSERTION REGION ONE HOMOLOG* (*BMI1B*) (Supplementary data 5). H2Aub mark was also gained in these genes (Supplementary data 5), although for most of the genes we did not observe transcriptional changes in *ubp5* under our analysed conditions. On the other hand, GO analyses of UBP5 target genes that gained the H2Aub mark in *ubp5* revealed a

significant over-representation of genes involved in response to DNA damage and repair (Supplementary Fig. 8d).

At the genome-wide level, UBP5 binding to chromatin typically occurs at the proximity of the transcription start site (TSS) and the start of the coding region (Supplementary Fig. 6a). Analyses of UBP5 binding peaks showed that majority of these sites correspond to protein coding genes, particularly exons and 5'UTRs that respectively correspond to ~51% and ~23% of the binding sites (Supplementary Fig. 9). Hence, we evaluated the impact of UBP5 in the transcriptional output of its target genes by integrating our ChIP-seq and RNA-seq data. We found a clear link between UBP5 gene binding and repression, as a significant ~62% (296/478) of the genes downregulated in *ubp5*

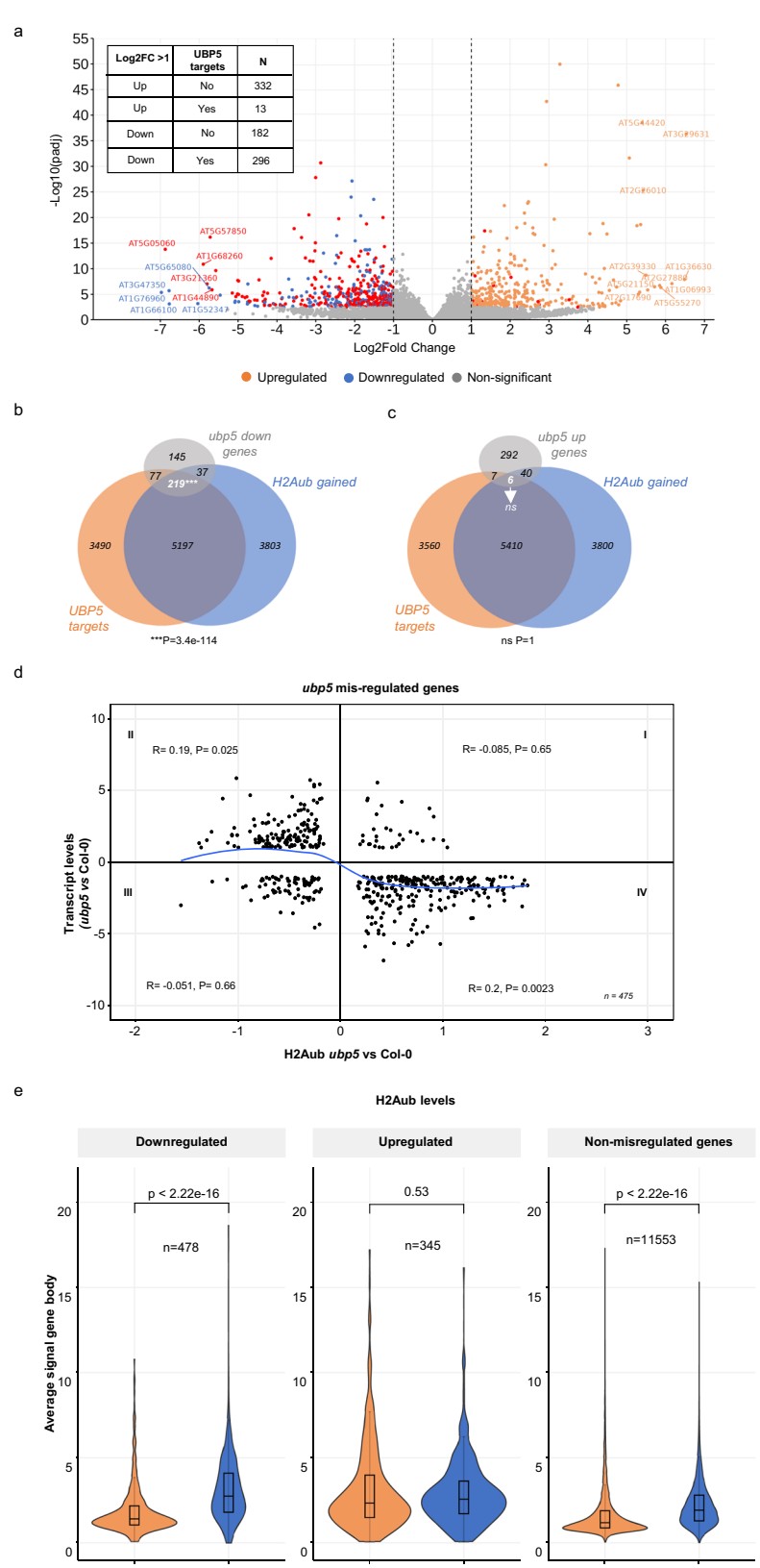

correspond to UBP5 targets, whereas only a few UBP5 target genes were upregulated (13/345 genes) (Fig. 4a). In addition, 52% of down-regulated genes in *ubp5* (219/478) were both UBP5 targets and gained H2Aub in the mutant, but these two conditions were almost never found associated to upregulated genes (Fig. 4b, c). More generally,

*ubp5* related defects in transcription are associated to an increase in H2Aub (Fig. 4d). Figure 4e shows that downregulated genes in *ubp5* have a significant probability to have increased H2Aub levels, while non-significant changes are observed in *ubp5* upregulated genes. Nevertheless, an increase in H2Aub does not always result in

**Fig. 4 | UBP5 mediates transcriptional de-repression. a** Volcano plot representing the UBP5 target genes which were downregulated and upregulated in *ubp5*. Up- and down-regulated DEGs are shown in orange and blue color dots respectively. Non-significant genes are shown as grey dots and UBP5 target genes as red dots. FDR < 0.05 and FDR values are derived from DESeq2 package by adjusting *p*-values using Benjamini-Hochberg method. **b** & **c** Venn diagrams showing overlap between UBP5 targets, H2Aub gained genes (hypermarked + de-novo) in *ubp5* and (**b**) downregulated or (**c**) upregulated genes in *ubp5* mutant. Super exact test was performed to check the statistical significance (*** represents significance with the *p* value = 3.4e-114, ns signifies non-significance). **d** Scatter plot showing the correlation between H2Aub and gene expression changes between Col-0 and *ubp5* plants. The x-axis shows Log2FC levels of H2Aub marked genes (FDR < 0.05) and y-axis shows expression in Log2FC of misregulated genes in *ubp5* as determined by DESeq2 (>1 fold variation, FDR < 0.05). For each quadrant, the correlation coefficient (R) along with the significance (*p*-values) are shown. The blue curve shows trend-line from LOWESS smoother function. Quadrant IV shows higher correlation between low expressed genes and hyper-marking of H2Aub. **e** Violin cum box plots represents the average signal of H2Aub at gene body for downregulated, upregulated genes and non-misregulated genes in Col-0 and *ubp5*. The violin plots show the distribution pattern of data and are overlaid with boxplots. For box and whiskers plots the middle line represents the median; the upper and lower lines are the first and third quartile (Q1 and Q3); the whiskers indicate the upper and lower limits of data spread by subtracting 1.5* interquartile range (IQR) from Q1 and adding 1.5* IQR toQ3. Statistical significance was calculated with two-sided Wilcoxon rank sum test, *p*-values are indicated above the plot.

transcriptional changes as a slight but still significant increase in H2Aub was generally observed in *ubp5*.

Taken together, our transcriptomic and epigenomic data suggest a role of UBP5 in relieving H2Aub-mediated repression, thereby promoting gene expression. Therefore, UBP5 seems to be predominantly involved in H2Aub erasure, which, at least for a significant set of its targets genes, results in transcriptional de-repression.

### UBP5-mediated H2A deubiquitination prevents deposition of H3K27me3

To explore whether UBP5 is targeted to chromatin in a sequence-specific manner, we analysed sequence motifs at UBP5 binding sites using MEME-ChIP[33] and identified a significant over-representation of GAGA and Telobox motifs (Fig. 5a). Notably, GAGA elements recognised by transcription activators/repressors and Telobox motifs typically recognised by TRBs are involved in recruiting PRC2[17,34,35]. These results thus suggest the existence of sequence-specific mechanisms commonly recruiting UBP5 and PRC activity.

Therefore, to further unravel the relationship between UBP5 function and PRC2 activity, we analysed H3K27me3 bulk level by WB analysis and identified a 70% increase in its abundance in *ubp5* (Fig. 5b). We conducted ChIP-seq to further determine the genome-wide effects of UBP5 on H3K27me3. Our data showed a high overlap of H3K27me3 marked genes in Col-0 seedlings with previously published data[36] (Supplementary Fig. 10a; Supplementary data 3). Differential analysis of our H3K27me3 genome-wide data revealed 2779 gaining the mark (i.e.hyper-marked) and 2349 H3K27me3 hypo-marked genes in *ubp5* (Supplementary Fig. 11a; Supplementary data 7). Notably, at UBP5 target genes average H3K27me3 level was higher in *ubp5* (Fig. 5c–e), while no increase was observed in non-UBP5 targets; instead, only a slight but statistically significant decrease was noted (Supplementary Fig. 11b, c). Further analyses of ChIP-seq data based on differential analysis showed that in 640 genes the following conditions concurred: i) H3K27me3 and ii) H2Aub gained in *upb5*, and iii) directly bound by UBP5 (Fig. 5f, g; Supplementary Fig. 10b), indicating that UBP5 not only erases H2Aub but also impairs the deposition of H3K27me3 at multiple sites. On the other hand, less than 4% of H3K27me3 hypo-marked genes were UBP5 targets (Supplementary Fig. 10c), suggesting that UBP5 may not play a direct role in H3K27me3 maintenance at these genes and therefore these changes might likely result from indirect effects in the regulation of H3K27me3 writers' or erasers' activity.

In agreement with a repressive role of H3K27me3 marking, average H3K27me3 levels in the gene body of *ubp5* downregulated genes was significantly higher than Col-0 levels, and there were no significant changes in the upregulated genes (Fig. 5h) and, similarly, there is a negative relationship between H3K27me3 and transcript levels in *ubp5* with almost all the down-regulated genes enriched with H3K27me3 (Fig. 5i). Nevertheless, as we have observed for H2Aub, gaining H3K27me3 does not always result in transcriptional changes as a subtle but significant H3K27me3 increase in non-misregulated *ubp5* genes was observed (Fig. 5h). In summary, the combined analysis of our transcriptomic and epigenomic data points towards the activity of UBP5 in de-repressing a set of its target genes by preventing H3K27me3 enrichment.

To understand how both H2Aub and H3K27me3 dynamics affect the transcriptional levels of genes we focussed on the set of genes which gained H2Aub in *ubp5*. In this set of genes, we analysed the transcriptional levels of H3K27me3/H2Aub marked genes in both Col-0 and *ubp5* and found that in both backgrounds, genes that are exclusively marked by H2Aub are more highly expressed than genes with the two marks or only H3K27me3, as previously shown (Zhou et al., 2017). On the other hand, while in Col-0 plants there is a significant difference in transcriptional levels of H2Aub/H3K27me3 versus H3K27me3 marked genes, this difference is lost in *ubp5* with both categories showing similar repressive levels (Supplementary Fig. 10d). Hence, UBP5 may contribute to pose H2Aub/H3K27me3 marked genes in a more responsive chromatin structure. Overall, we thus conclude that in the subset of 640 genes, UBP5-mediated H2Aub deubiquitination prevents the deposition of H3K27me3 mark leading to a de-repressed chromatin environment (Fig. 6).

## Discussion

PRC2 interactors play a key role in regulating its molecular activities and recruitment to chromatin[7]. For instance, we previously showed that PWO1 may mediate in providing PRC2 with the right chromatin environment to methylate H3[14]. In addition, PWO1 was proposed to form part of the PEAT complex[15]. Therefore, unravelling the protein interactors associated with epigenetic pathways can provide important clues to understand their possible crosstalk and activities. UBP5 was identified co-immunoprecipitating with all main components of PEAT[15] and, more recently, in a paper by Zheng et al. that was published during the revision of our manuscript, its role as a PEAT component has been confirmed. Notably, UBP5 molecular activity depends on its interaction to PWO1 and nucleosomal DNA[37]. Here, we have shown that co-expression of *PWO1* and *UBP5* in *N. benthamiana* tethers UBP5 to nuclear speckles. Hence, it is tempting to speculate that PWO1-UBP5 interaction is required for its recruitment to specific chromatin regions. Furthermore, our results indicate that UBP5 and PWO1, two components of the PEAT complex, may also interact with PRC2[14]. In addition, UBP5 binding sites are enriched in Telobox motifs previously involved in PRC2 recruitment to the chromatin by TRBs[17]. Strikingly, TRBs also form part of PEAT[15]. Considering the evidence, a very intriguing hypothesis is that PRC2 and PEAT coordinate their activities to dynamically regulate chromatin (Fig. 6). Nevertheless, while the PWO1-UBP5 interaction was observed in plant extracts[18] and has now been confirmed by our protein-protein interaction analyses, the UBP5-PRC2 binding has so far only been tested through heterologous systems (i.e. Y2H and co-IP in *N. benthamiana*). Hence, future *in planta* analyses will aid to confirm this interaction and to disentangle the complexity of UBP5's interacting network.

UBP5 belongs to the UBP family, which is part of the conserved DUB superfamily. Several DUBs are involved in the regulation of chromatin and some of them especially in H2A deubiquitination[11]. For instance, Drosophila protein Calypso as well as its corresponding ortholog in humans, the tumour suppressor BRCA-1-associated protein 1 (BAP1), form part of a PR-DUB complex able to remove the H2AK119ub1 mark. Intriguingly, PR-DUB has been described as a type of PRC despite its opposite activity to PRC1. Therefore, it seems that a dynamic ubiquitination/deubiquitination counterbalance is key for maintaining PRCs' activities and proper H2A ubiquitination levels over the genome[38–40]. The only proteins that have been related to H2A deubiquitination in Arabidopsis are the closely related UBP12 and 13 proteins, which were identified interacting with LHP1[12], a protein that may act as an accessory protein in both PRC2 and PRC1[7]. UBP12 was shown to be involved in the repression of a subset of PRC2 targets mediating H3K27me3 deposition and to be actively involved in H2A deubiquitination[12]. However, UBP5 has a much broader impact on H2Aub than the one so far described for UBP12/13 (Supplementary Fig. 12). Furthermore, UBP12/13-mediated H2Aub removal prevents loss of H3K27me3[13]. In contrast, our data indicate a role of UBP5 in preventing H3K27me3 gain at specific loci (Fig. 6). Finally, the genes that are regulated by UBP12/13 (i.e. H2Aub gained genes in *ubp12/13*) and UBP5 direct targets show little overlap (Supplementary Fig. 12), suggesting that they act through independent mechanisms or at different genome domains. However, as UBP12/13 direct target genes have not been described so far, this conclusion needs to be cautiously considered as indirect results in *ubp12/13* epigenomic data cannot be discarded[13].

UBP5 closest human orthologs are USP4, USP11 and USP15[11]. USP11 acts in both H2AK119 and H2BK120 deubiquitination and specifically deubiquitinates the histone variant γH2AX, which is key in homologous recombination[41], although no epigenomic data is available for the activity of these human proteins. Our ChIP-seq profiling in seedlings identified that UBP5 is required for H2Aub deubiquitination at a majority of PRC1-regulated Arabidopsis genes, and, considering *ubp5* phenotypes, UBP5 may have additional effects on H2Aub epigenome at other developmental stages. H2Aub ChIP-seq profile also points to a dual role of UBP5 deubiquitination activity. In ~20% of genes showing a H2Aub gain in *ubp5*, UBP5 acts to maintain a certain level of H2Aub in the plant; while, in ~40% of this set of genes, UBP5 fully erases this histone mark (Fig. 3d). Overall, these results indicate that UBP5 acts in *cis* to maintain the right H2Aub level at target genes with two possible scenarios for each locus: this modification is either 1) erased by UBP5 in most cells and therefore not detected in Col-0 plants but only in *ubp5* (i.e. de novo marked genes) or 2) stably present in Col-0 seedlings but removed by UBP5 only in certain genome copies or in certain cells (i.e. H2Aub hyper-marked genes). Further studies will be required to fully understand how UBP5 discerns between these different scenarios. Most UBP5 targets gained H2Aub in *ubp5* plants (i.e., 60% of UBP5 targets). However, a striking number of non-UBP5 targets were hypomethylated suggesting that UBP5 also indirectly reshapes the nuclear space. Understanding direct and indirect UBP5's impact on chromatin may help to discover new molecular mechanisms controlling epigenetic regulation.

Mirroring the meta-gene pattern of H2Aub in Arabidopsis[36] (Fig. 3d), UBP5 predominantly binds to chromatin in the vicinity of TSSs and at the start of protein coding regions. Furthermore, our transcriptional analyses show that UBP5 target genes tend to be downregulated in the *ubp5* mutant. These results point to UBP5 acting as a transcriptional activator, as shown for H2A deubiquitination in animals[42]. As UBP5 acts in histone deubiquitination, we favour the possibility of its active role in promoting transcriptional de-repression through the erasure of H2Aub as it has been proposed for other erasers (e.g. histone demethylases[43]). However, gain of H2Aub in *ubp5* is not always synonymous of changes in transcription in a comparable way as accessible chromatin is not always leading to activation[44].

The relationship between H2A and H3K27me3 deposition has been addressed in plants and metazoans[45], but much less is known about any possible crosstalk between the removal of these two marks. In line with the UBP5-PRC2 protein interaction identified here, UBP5 influences H3K27me3 levels at a majority of H3K27me3-marked genes (i.e., 5128 genes out of 7600 genes), ~20% of them corresponding to direct UBP5 target sites at the seedling stage (i.e., 1028 genes out of 5128 genes). For these genes, deposition of H2Aub plausibly precedes H3K27 trimethylation on the same nucleosome, as suggested for several PRC1/PRC2 target genes[46], and hence UBP5-mediated H2A deubiquitination will prevent H3K27me3 deposition by PRC2 (Fig. 6), probably making chromatin more accessible in these loci and leading to de-repression. Although further investigation will be required to fully address the UBP5-PRC2 relationship, a sequential mechanism in which the putative interaction between UBP5 and SWN is needed for 1) removal of H2Aub and 2) prevention of H3K27me3 deposition could explain how in a set of PRC2 target genes the removal of both marks is coordinated (Fig. 6). If prevention of H3K27me3 deposition occurs through UBP5-direct impairment of SWN activity/recruitment/stability and/or requires the activity of demethylases are attractive hypotheses to test in the future. Our proposed functional model also fits well with evolutionary results linking the deposition of H3K27me3 to the ubiquitination of H2A in *March-antia polymorpha*[47]. Despite all our results suggest an UBP5-PRC2 interaction, we should not forget that many UBP5 target genes that are enriched in H2Aub do not gain H3K27me3, indicating that UBP5 plays PRC2-independent functions uncoupling the removal of the two marks. This supports the possibility that PEAT can carry diverse activities or that different versions of PEAT can co-exist in the plant for a versatile chromatin regulation. In fact, Zheng et al. recently showed that PEAT is required for both H2A deubiquitination by UBP5, confirming our results, and H4K5 acetylation by HISTONE ACETYLTRANSFERASE OF THE MYST FAMILY (HAM) proteins[37]. Here, in addition to its H2A deubiquitination activity, we have demonstrated that UBP5 is able to antagonistically regulate H3K27me3 deposition and may interact with PRC2 components. Therefore, these different results may indicate the multi-functionality of PEAT to activate transcription. It also opens further fascinating questions about UBP5 alternative activities in the regulation of chromatin accessibility that we look forward to answering in future studies.

## Methods

### Plant materials and cultivation conditions

All *Arabidopsis thaliana* (Arabidopsis) lines used in this study were in the Columbia-0 (Col-0) ecotype background. For the generation of *ubp5* CRISPR-Cas9 mutant, double guide system of Cas9-directed mutagenesis was performed[48] to delete a fragment size of 3361 bp from *UBP5* gDNA sequence (Supplementary Fig. 2A). sgRNAs were designed using CRISPR-P tool[49]. The P3-Cas9-mCherry vector for generating the *ubp5* line was kindly provided by Charles Spillane[48]. Deletion of the genomic fragment from *UBP5* was confirmed using Sanger sequencing (LGC genomics, Germany). Transgenic plants were developed by *Agrobacterium*-mediated gene transformation with floral dip method[50]. For genotyping, DNA extraction was done based on this described protocol[51]. Oligonucleotide primers used for CRISPR-Cas9 mutagenesis and genotyping are indicated in Supplementary Table 1. For the *UBP5pro::gUBP5-GFP;ubp5* line, a 1708-kb-upstream fragment and gene-body regions of *UBP5* without stop codon were amplified from genomic DNA of Col-0 with GW-compatible primers (Supplementary Table 1). *UBP5pro::gUBP5* was fused with a C-terminal GFP sequence in the (pGKGWG) vector[52].

Sterilised seeds were sown on Murashige & Skoog medium (MS Base) supplemented with 1% Sucrose, 0.1% MES, 0.8% agar with pH

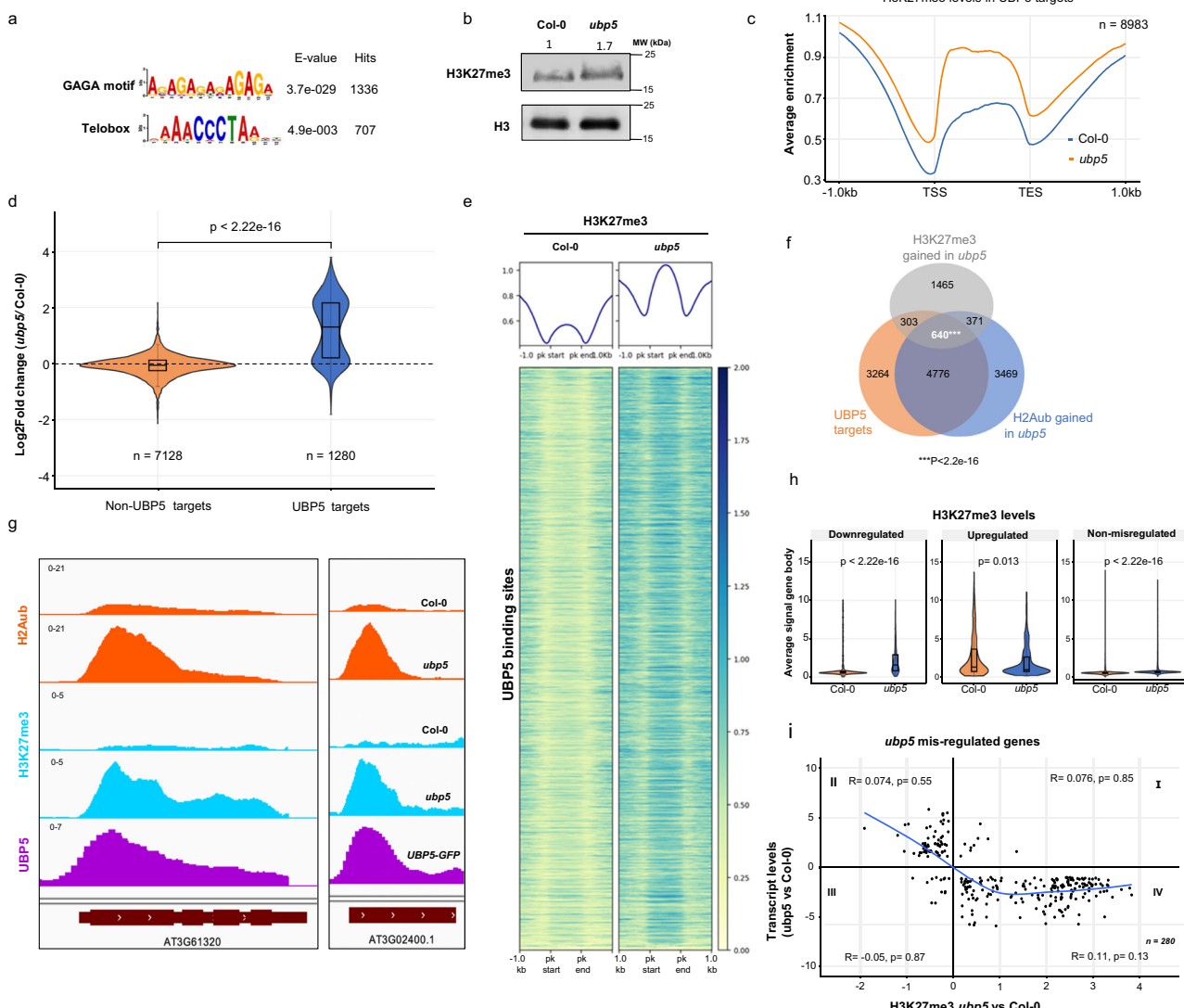

**Fig. 5 | UBP5-mediated H2Aub deubiquitination prevents deposition of H3K27me3. a** Motif enrichment analysis of UBP5 target genes. The sequence logos, accuracies and hits of the best motifs found by MEME-ChIP. **b** H3K27me3 levels in *ubp5* seedlings, Histone H3 used as a loading control. Numbers above blots indicate relative H3K27me3 levels. Three independent experiments were performed. **c** Metagene plot of average H3K27me3 enrichment over the UBP5 target genes. **d** H3K27me3 ChIP seq signal in *ubp5*/Col-0 between UBP5 targets and non-UBP5 targets. Violin overlaid boxplots display median (middle line), first and third quartiles (Q1 and Q3), with whiskers indicating data spread (1.5*IQR from Q1 and Q3). Statistical significance was determined by two-sided Wilcoxon rank sum test. **e** Heatmap showing the distribution of H3K27me3 on UBP5 binding sites for Col-0 and *ubp5*. UBP5 binding peaks are clustered based on higher to lower enrichment (top to bottom). **f** Venn diagram representing the overlap between UBP5 targets, H2Aub and H3K27me3 gained genes in *ubp5* (FDR < 0.05). Super exact test is

performed to test the overlap (*** represents significant overlap with *p* < 2.2e-16). **g** IGV browser snapshots of representative UBP5 target genes in which H2Aub and H3K27me3 are gained in the *ubp5* mutant. **h** Average signal of H3K27me3 at gene body for downregulated, upregulated and non-misregulated genes. Median (middle line), first and third quartiles (Q1 and Q3), with whiskers indicating data spread (1.5*IQR from Q1 and Q3). Statistical significance was determined by two-sided Wilcoxon rank sum test. **i** Scatter plot shows correspondence between H3K27me3 levels and gene expression changes between Col-0 and *ubp5* plants. The x-axis shows Log2FC levels of H3K27me3 marked genes by DESeq2 analysis (FDR < 0.05), y-axis shows expression of misregulated genes in *ubp5* (Log2FC > 1, FDR < 0.05). Blue curve shows trend-line from LOWESS smoother function, with Pearson correlation coefficient (R) and significance (*p* values) displayed. Quadrant IV shows significant enrichment of low expressed genes that have gained H3K27me3.

adjusted to 5.6, stratified at 4 °C for three days and placed in a Percival tissue culture cabinet under a 16:8 h light: dark (21 °C/18 °C) regime until they were transferred to soil. Arabidopsis plants were grown on pots containing compost, vermiculite and perlite (5:1:1 proportion) with the same photoperiod under fluorescent lamps at 200 µmol m⁻² s⁻¹. For hypocotyl and root length measurements, Col-0 and *ubp5* seeds were sown on MS medium, and the plates were placed vertically in the growth chamber in LD conditions. Photographs were taken at the end of 10 days, hypocotyl and root length were measured using the Fiji image processing software.

## Yeast two hybrid assay

For yeast two hybrid assays, untransformed *Saccharomyces cerevisiae* AH109 cultures were grown at 28 °C, on solid or liquid Yeast Peptone Dextrose (YPD) media supplemented with adenine (80 mg/L). The *S. cerevisiae* AH109 competent cells were obtained as previously described[53]. Yeast was co-transformed using a heat shock method at 42 °C for 30 min[54]. For plating, 3 µL of culture were plated at the same concentration on drop-out media (minimal medium) in the absence of leucine and tryptophan (SD-L-W) or more restrictive media without histidine as well (SD-L-W-H) in

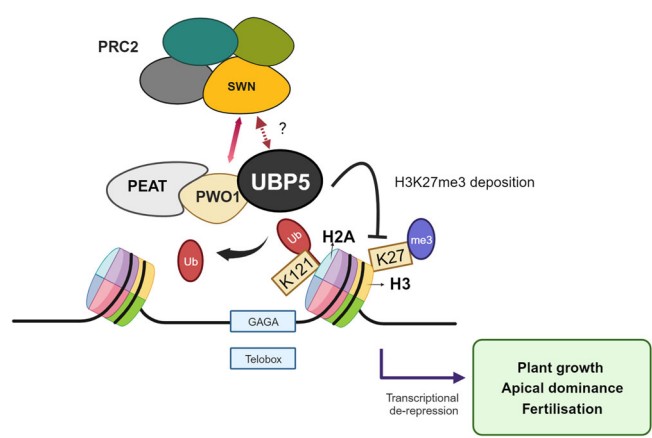

**Fig. 6 | Working model for UBP5 function.** UBP5 interacts with PWO1 (a PEAT complex subunit) and possibly with PRC2 through a direct interaction with SWN. UBP5 recruitment to chromatin associates with PRC2-related *cis*-elements (light blue boxes). UBP5 acts as a H2A deubiquitinase and prevents the deposition of H3K27me3, leading to transcriptional de-repression and changes in plant development. Created with BioRender.com.

serial dilutions. Yeast colony growth was analysed after 3 to 4 days growing at 28 °C. Both bait and prey empty vectors were used as negative controls.

## Co-immunoprecipitation assay
Modified versions of pMDC7 carrying the GFP or mCherry tags[55] were used to insert the coding sequence of *UBP5* and *SWNΔSET* via Gateway cloning (Invitrogen). Vectors were transformed in *Agrobacterium tumefaciens* (Agrobacterium) GV3101 pMP90. For transient expression assays, the abaxial sides of leaves of 4/5-week-old *N. benthamiana* plants were infiltrated with transformed Agrobacterium cell culture suspension in log phase growth. Expression was induced by spraying 20 μM β-estradiol in 0.1% Tween onto infiltrated leaves 48 to 72 h after Agrobacterium infiltration. Fluorescence was monitored in leaf epidermis cells after a short induction period (4–6 h when fluorescence was visible) using an Olympus BX51 epifluorescence microscope. After 6 h from the second induction of β-estradiol, the samples were frozen in liquid $N_2$. The samples were ground in a liquid $N_2$ precooled mortar followed by 20 min at 4 °C in a shaker in 10 mL of protein extraction buffer (10% glycerol, 150 mM NaCl, 2.5 mM EDTA, 20 mM Tris-HCl pH 8, 1% Triton and Complete® EDTA-free protease inhibitor cocktail (1 tablet/50 mL; Roche)). After resuspension, samples were filtered through two Miracloth (Calbiochem®) layers and centrifuge at 4 °C for 15 min at 4000 rpm. After centrifugation, the supernatants were transferred to a new 15 mL tube, and the extracts were taken, mixed with 3X Laemmli buffer (0.3 M Tris-HCl (pH 6.8); 10 % (w/v) SDS; 30% (v/v) glycerol; 0.6 M DTT; 0.01% (w/v) bromophenol blue) and heated at 95 °C for 5 min. Co-IPs were carried out by incubating the samples with 30 μL of protein A agarose bead slurry for 4 h at 4 °C in a rotating wheel and with anti-mCherry (Takara 632496, Dilution-1:1,000). After 4 h incubation, a centrifugation at 4 °C at 500 g for 2 min was carried out to precipitate the beads. The beads were washed 3 times with protein extraction buffer, resuspended in 3× Laemmli buffer and denatured at 95 °C for 10 min. Proteins were loaded in 10% SDS-PAGE gels and transferred to a PVDF membrane. Membranes were developed with anti-GFP (Roche 11814460001, Dilution-1:1000).

## Subnuclear localisation and FRET assay
For subnuclear localization in *N. benthamiana*, estradiol-inducible pMDC7-derivatives plasmid vectors containing our coding sequences were transformed into Agrobacterium (GV3101 PMP90 strain with

p19 silencing suppressor plasmid). FRET-APB assays were performed in *N. benthamiana* Agrobacterium-infiltrated with the corresponding estradiol-inducible pMDC7 vectors carrying the appropriate cDNAs. After infiltration (following the same protocol as described for co-IPs) and after 16-20 h after β-estradiol induction, leaf pieces were observed under a LSM780 (Zeiss) by excitation at 488 nm (argon laser) for eGFP and 551 nm (helium laser) for mCherry tagged proteins. FRET efficiency was calculated as the intensity increase in GFP fluorescence signal after photobleaching of the acceptor mCherry only using measurements with less than 10% GFP intensity fluctuations before acceptor bleaching[55].

## Histone extraction and western blot
Nuclei were extracted from 1.5 g of 12 days after germination (DAG) seedlings using nuclei extraction buffer (0.4 M Sucrose, 10 mM Tris-HCl pH 8.0, 5 mM β-Mercaptoethanol, 10 mM $MgCl_2$, 0.1 mM PMSF). Extracted nuclei were treated overnight with 0.4 N $H_2SO_4$ to obtain a histone-enriched extract. The extracted proteins were precipitated with 33% trichloroacetic acid and then washed 3 times with acetone, air-dried, and re-suspended in 100 μL 3X Laemmli buffer. The samples were boiled for 10 min, separated on 15% sodium dodecyl sulfate-polyacrylamide electrophoresis gels and transferred to a polyvinylidene difluoride membrane (Immobilon-P Transfer membrane, Millipore) by wet blotting in transfer buffer (25 mM Tris–HCl, 192 mM glycine, and 10% methanol). Primary and secondary antibodies used were anti-H2Aub antibody (Cell Signalling Technology D27C4, dilution- 1:2000), anti-H2A antibody (Active Motif 91325, dilution- 1:1000), anti-H3K27me3 antibody (Millipore 07-449, dilution- 1:5000), anti-H3 (Abcam ab1791, dilution- 1:5000), anti-mouse IgG (H+L) HRP conjugated (Chemicon International AP308P, dilution- 1:3000) and Anti-Rabbit IgG (whole molecule)–Peroxidase (Sigma Aldrich A9169, dilution- 1:63000). Chemiluminescence detection was done with Super-Signal West Pico or Femto (Thermo Fischer Scientific) following the manufacturer's instructions.

## ChIP-qPCR, ChIP-seq and data analyses
Chromatin immunoprecipitations (ChIP) were carried out using 12-DAG seedlings. Chromatin was extracted from formaldehyde fixed tissue and fragmented using a Bioruptor® Pico (Diagenode) in fragments of 200–500 bp. Antibodies used for ChIP-qPCR in this study were H3K27me3 (Millipore 07-449) and H2Aub (Cell Signalling Technology D27C4). 30 μL/sample of Protein A Dynabeads (10002D) were used for preclearing before IP. The IP was performed with 60 μL/sample of Protein A Dynabeads and 5 μL of antibodies in the ChIP dilution buffer at 4 °C overnight. Following IP, chromatin was washed with four different wash buffers- Low Salt, High salt, LiCl and TE wash buffer sequentially. Then, the chromatin was eluted, and crosslinking was reversed overnight at 65 °C. After IP, DNA was eluted and purified using ultrapure phenol:chloroform:isoamyl alcohol (25:24:1) pH 8.05 followed by ethanol precipitation. Input DNA was diluted to 1:10, and 1 μL of IP DNA was used for quantitative PCR (qPCR). ChIP-qPCRs were carried out in a CFX96TM Real-Time PCR Detection System (Bio Rad) using TakyonTM No Rox SYBR MasterMix dTTP Blue (Eurogentec).

For ChIP-seq experiments, chromatin extraction and immunoprecipitation of histones were done in three biological replicates for H2Aub and two biological replicates for H3K27me3 at 12-DAG-old Col-0 and *ubp5* seedlings. Two IPs were carried out for each biological replicate using 100 μg of chromatin, quantified using Pierce BiCinchoninic Acid (BCA) assay kit (Thermo Fisher Scientific). After IP, DNA was eluted and purified. Library preparation and paired end sequencing was performed using DNA Nanoballs (DNB™) sequencing technology from BGI (Sequencing method: DNBSEQ-G400_PE100). Reads were mapped using STAR v2.7.8a[56] onto TAIR10 Arabidopsis with parameters align intron max as 1 and align ends type as EndToEnd. The organelle genomes were excluded from the mapped reads. Duplicated

reads were removed using Picard tool MarkDuplicates option. Only uniquely mapped reads were retained for further analysis. Marked peaks for each IP were obtained using MACS3[57] with parameters broad peak and q value cut off as 0.05. Browser tracks were obtained using the bamCoverage function by scaling with the parameter --normalizeUsing RPGC. Tracks were visualised using IGV v2.12.3[58]. Bedtools Utility Intersect[59] was used to intersect the MACS3 peaks obtained from the biological replicates. The resulting peaks from the biological replicates were merged and annotated with TAIR10 gene coordinates. To determine gain or depletion of H2Aub or H3K27me3 marks, the number of reads mapping into the peak coordinates was calculated using Bedtools Utility Multicov and the peaks from all samples were grouped by gene-ID to obtain unique peak coordinates per marked gene using Bedtools Utility Groupby v2.26.0[59]. Differential enrichment of respective marks between samples were done using DESeq2 analysis[60]. The comparison between biological replicates of H2Aub and H3K27me3 are shown in Supplementary Fig. 13a, b.

## UBP5-GFP ChIP-seq and data analyses

UBP5-GFP ChIP was performed with the *UBP5pro::gUBP5-GFP;ubp5* line using a double crosslinking protocol[61]. Two biological replicates with 2 g each from 12-DAG seedlings were ground in liquid $N_2$ to fine powder and resuspended in nuclei isolation buffer (60 mM HEPES pH 8.0, 1 M Sucrose, 5 mM KCl, 5 mM $MgCl_2$, 5 mM EDTA, 0.6% Triton X-100, 0.4 mM PMSF, pepstatin and complete protease inhibitors (Roche)). Then, the samples were cross-linked with 25 mM ethylene glycol bis succinimidyl succinate (EGS) by rotating for 20 min and with 1% formaldehyde by rotating for 10 min. The crosslinking of samples was stopped by 2 M glycine for 10 min at room temperature. The chromatin was isolated and sheared into 200–500 bp fragments by sonication. For IP, the sonicated chromatin was incubated with 20 µl of anti-GFP antibody (Thermo Fisher #A11122) overnight at 4°C while gentle rotating. Followed by IP, eluted and purified DNA of two independent biological replicates along with input control without antibody was used for library preparation and paired end sequencing was performed using DNB[TM] sequencing technology from BGI.

For UBP5-GFP ChIP-seq data analysis, raw data with adapter sequences or low-quality sequences was filtered using SOAPnuke software (BGI). The reads were mapped to the Arabidopsis genome (TAIR10) using Bowtie2 2.4.5[62] with default parameters. Only uniquely mapped reads were retained for further analysis. Peaks were called using MACS3[57]. The peaks were converted to bigwig files using deepTools[63]. bamCoverage was done using RPGC normalisation. The intersections of common peaks between two biological replicates with FDR < 0.01 was obtained using Bedtools Utility Intersect v2.30.0[59]. Comparison between ChIP-seq replicates were shown in Supplementary Fig. 13a, b.

For DNA motifs analyses, we considered -500 bp to +250 bp from TSS for the UBP5 target genes using 'getfasta' function. We searched for enriched DNA motifs using the fasta file as a input for MEME-ChIP[33] with discriminative mode using the negative control sequences wherein UBP5 targeting regions were removed.

## H2A deubiquitination assay

*N. benthamiana* plants were infiltrated with Agrobacterium strain GV3101 carrying appropriate binary vectors. Before infiltration, bacterial cells were pelleted and resuspended in MES buffer (10 mM $MgCl_2$, 10 mM MES, 200 µM acetosyringone, pH 5.7) in the dark for 2 h at room temperature (RT). The suspended Agrobacterium cells were mixed with P19 silencing suppressor at an appropriate ratio to a final $OD_{600}$ of 0.6, followed by infiltration in 6-week-old *N. benthamiana* leaves. After 24 h of infiltration, an induction was performed by spraying the leaves on their abaxial side with 50 µM of β-estradiol solution in 0.1% (v/v) Tween-20. Post induction, the transient expression of agroinfiltrated cells was examined under an Olympus BX51

epifluorescence microscope. Histone extraction was performed from the infiltrated leaves and immunoblotted using anti-H2Aub antibody (Cell Signalling Technology D27C4, dilution- 1:2000), and anti-H2A (Active Motif 91325, dilution- 1:1000) and anti-H3 (Abcam ab1791, dilution- 1:5000) as controls.

## RNA isolation, quantitative RT PCR

Total RNA was isolated from 12-DAG seedlings (Col and *ubp5*) using E.Z.N.A. Plant RNA Kit (OMEGA biotek) following manufacturer instructions. The RNA concentration was determined using the Nanophotometer (IMPLEN). RNA was examined by electrophoresis on a 1.2% agarose gel. For cDNA synthesis, RNA samples were subjected to DNAse treatment and cDNA synthesis was performed using (Thermo Scientific). Quantitative real time PCR (qRT-PCR) was performed in a CFX96[TM] Real-Time PCR Detection System (Bio Rad) using TakyonTM No Rox SYBR MasterMix dTTP Blue (Eurogentec). Expression levels were normalised to the reference genes At5G25760 and At4G34270[64]. Relative enrichment was calculated using the $2^{-\Delta\Delta CT}$ method[65].

## RNA-seq library preparation, sequencing, and bioinformatics

For RNA-seq, RNA was extracted from 12-DAG seedlings with four biological replicates for each background (Col-0 and *ubp5*). Library preparation and RNA-seq was performed according to the protocol described recently[66]. 500 ng DNase-treated RNA was used for reverse transcription with 50 mM different barcoded oligo(dT) primers and SuperScript III (Thermo Fisher Scientific). Each reaction was pooled, pools were Ampure purified (1.5x beads to sample volumes) and then eluted. Second-strand synthesis was carried out using nick translation protocol (Krzyszton et al. 2022). Tagmentation reaction[67] was performed using recovered dsDNA sample incubated with homemade Tn5 enzyme in a freshly prepared 2x buffer (20 mM Tris-HCl pH 7.5, 20 mM MgCl2, 50% DMF). Illumina indexing PCR was performed using the tagmented DNA. Libraries were sequenced on Illumina NextSeq 500 system using the paired-end mode to obtain 21 nt R1 (contain barcode and Unique Molecular Identifier (UMI)) and 55 nt R2 (contain mRNA sequences).

After quality control using fastqc, reads R1 and R2 were processed separately. In our oligo(dT) primers two parts of UMI are split by barcode sequence, therefore we transformed read R1 fastq file using awk command. Read R2 was trimmed to remove potential contamination with poly(A) tail using BRBseqTools v 1.6 Trim[68]. Reads were mapped using STAR v 2.7.8a[56] to TAIR 10 genome with Araport11 genome annotation. Finally, the count matrix for each library and each gene was obtained using BRBseqTools (v 1.6) CreateDGEMatrix[68] with parameters *-p UB -UMI 14 -s yes*, using Araport11 genome annotation and a list of barcodes. The differential gene expression analysis was done using the DESeq2[69]. Further, the genes were filtered based on log2 fold-change of ±1 and an adjusted *p*-value of less than 0.05 and categorised as upregulated, downregulated, and non-misregulated genes. GO enrichment analyses were performed in different gene set using ShinyGO tool v0.75[70].

## Statistics and reproducibility

Statistical tests performed on experimental data and sample sizes are noted in figure legends. No statistical method was used to predetermine sample size. All data points are derived from biological replicates. Exact *p*-values for each pairwise comparisons are mentioned in the figures. Plants were placed randomly in the plant growth facility. No data were excluded from the analyses. No blinding was applied for sampling.

## Reporting summary

Further information on research design is available in the Nature Portfolio Reporting Summary linked to this article.

## Data availability

All data supporting the findings of this study are available within the manuscript and its supplementary files. All high-throughput sequencing data generated in this study have been deposited in GEO with accessions codes GSE217614 and GSE209707 and data lists are available in the Source Data file. Seeds of the *ubp5* CRISPR/Cas9 line are available under request. Source data are provided with this paper.

## Code availability

The custom code used for the analysis has been deposited at [https://github.com/mohang13/ubp5_nat_comms]. Any additional information required to reanalyse the data reported in this paper is available from the corresponding author upon request.

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

## Acknowledgements

S.F. acknowledges support from the College of Science and Engineering and the Research Office (University of Galway). S.F. is grateful to Jennifer Siobal for technical support. S.F. thanks Miguel de Lucas Torres (Durham University) for his support with Y2H experiments. S.F., R.H., K.N. and M.G. were supported by 20/FFP-P/8693 grant from Science Foundation Ireland (SFI) and by a NUI Galway Research Grant for Returning Academic Careers QA151. M.G. was also supported by the SFI Centre for Research Training in Genomics Data Science (grant N. 18/CRT/6214). J.G. was supported through the University of Galway Hardiman Scholarship programme and Thomas Crawford Research Grant. J.G. internship at IBENS was supported by the COST Action CA16212 INDEPTH (EU). E.M. was funded by a College of Science and Engineering scholarship (University of Galway). Work in FB and CB laboratory was supported by ANR-18-CE13-0004-01 and ANR-20-CE13-0028 grants from the French National Research Agency. S.S. was supported by Foundation for Polish Science (TEAM POIR.04.04.00-00-3C97/16) and by Polish National Science Centre (SONATA BIS UMO-2018/30/E/NZ1/00354). M.K. was supported by Polish National Science Centre (OPUS UMO-2021/41/B/NZ3/02605).

## Author contributions

J.G. and S.F. conceptualised the experiment approach and designed the methodology; J.G., K.N., E.M., L.W., R.H. and J.L. performed the experiments; J.G., M.G., A.F., M.K., F.B. and C.B. performed the genomic data curation, analysis, and visualisation; J.G. and S.F. wrote the original manuscript; J.G., F.B., C.B., S.S., D.S. and S.F. contributed to the interpretation of results; all the authors contributed to manuscript revision and approved the final manuscript.

## Competing interests

The authors declare no competing interests.
