## [Peer Review File · Nature Communications]

The UBP5 histone H2A deubiquitinase counteracts PRCs-mediated repression to regulate Arabidopsis developmentREVIEWER COMMENTS

Reviewer #1 (Remarks to the Author):

Godwin et al. discovered UBP5, one previously identified PRC2-interacting protein, as histone H2A deubiquitinase. The H2Aub level is increased in *ubp5* mutant. The authors also reported increased H3K27me3 in *ubp5* mutant, and concluded that UBP5-mediated H2Aub is deubiquitination prevents H3K27me3 deposition. Linking H2Aub mediated by PRC1 and H3K27me3 mediated by PRC2 is an interesting issue. However, I have several major concerns about the data analysis and interpretation.

1. The underlying mechanism is unclear. SWN is the catalytic subunit of PRC2, responsible for depositing H3K27me3. While UBP5, which interacts with SWN, prevents H3K27me3 deposition. Whether the prevention of H3K27me3 deposition by UBP5 is independent of SWN interaction?
2. There are three major conclusions, i) UBP5 is responsible for H2Aub, which seems apparent from the data (despite only the overlap is presented without statistical test); ii) UBP5 prevents H3K27me3 deposition. This is weakly supported by the data without statistical test. Whether UBP5 overexpression would lead to H3K27me3 increase? iii) UBP5 counteract with PRC2 repression. This is not supported by the transcriptomic data in Fig. 5H. No clear relationship could be observed between H3K27me3 increase and expression decrease.
3. The major role of PRC1 is catalyzing H2A ubiquitination, and the effect of PRC2 is recognizing H2Aub and catalyzing H3K27me3. Previous reports indicated that H3K27me3 and H2Aub overlap is limited. The identification of UBP5 responsible for removing H2Aub may help explain this phenomenon. One possible mechanism may be that the interaction of UBP5 and SWN is to remove H2Aub and facilitate H3K27me3 at certain loci. Given the genome-wide influence of PcGs, it is difficult to be explained by a single model in Fig. 6. The authors may perform genetic experiment to elucidate the genetic interaction between PRC1 and UBP5, in order to see if UBP5 is antagonistic of the effect of PRC1. Focusing on the interaction between H2Aub and deubiquitination seems more direct and straightforward.
4. The title need to be re-considered, which indicates UBP5 is associated with stress responses, but merely based on GO enrichment analysis without experimental validation.
5. Line20, "there is limited knowledge about PRCs' interacting proteins and their interplay with PRCs in epigenome reshaping, which is fundamental to understand gene regulatory mechanisms". There are already a large number of PRC interacting proteins reported, as reviewed by authors of this manuscript (10.3390/epigenomes6010008).
6. line 121 shows that *UBP5pro::UBP5-eGFP* can restore the phenotype, and the subsequent analysis lacks the data of *UBP5pro::UBP5-eGFP*.

Other comments:

1. To demonstrate the reproducibility of replicates, data from WT and mutants need to be brought together. WT and mutant data can also correlate well if only pairwise comparisons are performed, as transcriptome and epigenomic data are largely unchanged between samples, with quantitative differences occurring at only a few hundred loci.
2. The conclusions of Fig. 4A-B and Page vi, line 192-194 based on proportion instead of statistical analysis, and can not answer whether UBP5 binding gene is significantly enriched in down-regulated expression genes. And the ratio of overlap should not only be compared with up-regulated genes, but also with all genes. The results of Fig4D also lack comparisons with genes without expression changes.
3. The conclusion in Page vii, line213-214 "Notably, high H3K27me3 level was particularly

pronounced at gene domains corresponding to UBP5 binding sites " lacks control, i.e., Fig5C-D lacks the situation in non UBP5 binding sites of the genome as a control, Fig5E, 5G, 5H also lack statistical analysis against the genome-wide background.

Reviewer #2 (Remarks to the Author):

The manuscript by Godwin et al. described the interaction between UBP5 histone H2A deubiquitinase and PRC2 components and the functions of UBP5 in Arabidopsis. The topic is interesting, but the data is too preliminary to get a conclusion and lack of novelty.

The major concerns:

1, the interaction between UBP5 and PRC2 is not solid. A, for FRET assay, usually the cutoff for FRET efficiency to determine the interaction between two proteins is 5%. However, the efficiencies for CLF-PWO1 and UBP5-PWO1 is less than 5%; B, for Y2H assay, 1, it is not clear for the condition (medium) for yeast growth in the test group (it shows in the Figure 1D that the medium is WLH lack, but it is indicated that a LWAH lack medium in the Figure legend), 2, both UBP5-BD and SWN Δ Set-AD are self-activated, thus it not suitable for Y2H assay; 3, it is not clear why use SET domain-deleted SWN but not full length SWN; 4, it is strange and unusual why both un-diluted UBP5-BD and un-diluted SWN-AD exhibited strong self-activation, while diluted UBP5-BD and diluted SWN-AD exhibited no self-activation; C, for Co-IP assay, 1, only detected the interaction between the two proteins under overexpression tobacco cells, not in planta Co-IP assay; 2, It is not sure that whether anti-GFP antibodies can distinguish GFP and mCherry or not; 3, the controls are missed in the Co-IP assay (loading control, express mCherry but not recognized by anti-GFP antibodies in both IP and Co-IP assays). In addition, the authors observed that UBP5 interacts with PWO1, SWN, CLF, EMF2, VRN2, FIE, MSI1, and BMI1B under their conditions. It is not clear why a predicted deubiquitinase interacts with so many proteins? Therefore, take all these in together, it can not rule out the possibility that the interaction between UBP5-PWO1 (and PRC2) is non-specific.

2, the data in Figure 2 does not support the conclusion that UBP5 is an essential plant stress response regulator. There is no stress phenotype observed in *ubp5* mutant, GO category does not fully support the stress response.

3, there is no direct evidence to support the conclusion that UBP5 acts as a H2A deubiquitinase. The data suggests that UBP5 is required for H2A deubiquitination, but no deubiquitinase activity for UBP5 is provided.

4, as H3K27me3 is dependent on H2Aub, therefore it is not surprising that UBP5-mediated H2Aub deubiquitination prevents deposition of H3K27me3.

Reviewer #3 (Remarks to the Author):

The manuscript by Godwin et al provides evidence of a new PRC2 binding partner that is responsible for deubiquitination of the PRC1 mark H2Aub in Arabidopsis. Phenotypic analysis of *ubp5* mutant demonstrates the role of UBP5 in regulation of Arabidopsis development. RNAseq experiment shows significant transcriptomic alterations in *ubp5* mutant compared to WT plants. In line with other UBP family members, UBP5 is involved in deubiquitination of histone H2A as revealed by western blot and H2AK121ub ChIPseq analysis. The authors propose a model whereby UBP5 protein binds to the 5' end of

Arabidopsis PRC target genes to relieve H2A-mediated repression thus promoting gene reactivation. At a number of PRC2 target genes, UBP5 may additionally prevent deposition of H3K27me3. Overall, this novel research delivers a new component taking part in PRC regulation in Arabidopsis. More specifically, the data presented supports the finding of a new eraser of H2Aub likely also affecting deposition of H3K27me3. I have some comments to improve this work.

Major points

1- Figs. 1A-B show UBP5 nuclear localisation in *N. benthamiana*. To improve this work, the authors could provide evidence of nuclear localisation in *A. thaliana* using the UBP5pro:UBP5-eGFP line generated in this work. Given the phenotypes of the *ubp5* mutant reported in Fig. 2A-B, I wonder whether UBP5-GFP expression is restricted to certain tissues of the plant (i.e. meristematic tissues). Fig. 1A could also be improved, it is not clear what the bright field is representing. I would suggest providing a wider view in *N. benthamiana* capturing several cells with UBP5 nuclear localisation.

2- The authors show interaction of UBP5 with the PRC2 component SWN. However, a mutated version of SWN lacking the SET domain has been used both in Y2H and coIP assays. To further support the interaction of UBP5 and SWN, I would suggest testing UBP5 interaction with full length SWN.

3- Related to the Y2H experiment (Fig. 1D and Suppl fig. 1), UBP5-BD has been used to test interaction with PWO-AD and SWN Δ SET-AD, whereas UBP5-AD has been tested against EMF2-BD. Please provide the reciprocal combinations for all interactions. In addition, please edit suppl. Fig. 1. I understand that the authors have included PWO1-SWN Δ SET and PWO1-UBP5 as positive controls for the EMF2-UBP5 interaction, but actually, these are exactly the same images as in Fig. 1D, not an independent set of positive controls.

4- The de-repression of PRC2-regulated genes such as STM, WUS and KNATs (Suppl. Fig. 4) correlates well with the phenotypes observed in the *ubp5* mutant (Fig. 2A). However, given the function proposed for UBP5 as a H2Aub deubiquitinase that prevents PRC2 activity (Fig. 6), de-repression of these developmental genes seem to be an indirect effect of absence of UBP5. I was wondering if the authors would be able to identify genes among the UBP5 targets, whose H2Aub and/or H3K27me3 levels are affected in the *ubp5* mutant, that may better explain the observed phenotypes.

5- This study proposes a direct role of UBP5 in H2Aub deubiquitination of numerous targets (Fig. 3B-H) thus having an effect on global levels of H2Aub and H3K27me3 (Fig. 3A and 3B). However, one could not rule out an indirect effect of UBP5 on the mentioned histone modifications. Since the authors count with an inducible i35S:UBP5-GFP construct, it would be interesting to test whether inducing UBP5 expression would have a global effect on H2Aub levels in the *N. benthamiana* system.

6- Regarding Fig. 3B, the description of the different categories presented in the Venn diagram could be improved. It would also be helpful to point here which are the "hyper-marked" and the "de novo-marked" genes. It is not clear to this reviewer what is the difference between the 1,818 and the 4,383 genes under the *ubp5* diagram.

7- The discussion section is quite long, covering aspects of the UBP5 orthologs in plants and

animals as well as the possible role of UBP5 in stress responses and in the PRC1/PRC2 dynamics. These are all indeed very exciting topics and it's worth mentioning them. Yet, I would suggest trying to focus more on linking these aspects to the phenotypes observed in the *ubp5* mutant.

Minor points

- Lines 74-77: The enrichment of GAGA and Telobox motifs over UBP5-GFP binding sites is indeed interesting and provides a good link to PRC2 regulation. However, referring to UBP5 as a “sequence-specific eraser” may not be appropriate. It sounds like UBP5 is able to bind DNA by itself.

- Line 112, “UBP5 is an essential plant developmental and stress response regulator”. Considering the data presented in this work, I agree that UBP5 is required for Arabidopsis development, but I am not convinced about its role in stress responses. The authors are not providing experiments to support UBP5 function in stress responses.

- According to the text (lines 106-107), SWN Δ SET was used in colP experiments. Please correct SWN-GFP in Fig. 1E.

- Line 573, Fig. 1 legend: please correct FRET-APB

- Legend of Suppl. Fig. 4: remove ns as it does not appear in any of the plots.

We would like to express our sincere gratitude to the Reviewers for their valuable input and constructive feedback, which has significantly contributed to the improvement of the manuscript. We have meticulously addressed the concerns raised by the Reviewers, addressing each point with thorough and detailed responses. Additionally, the edited sections in the manuscript have been highlighted in yellow for the Reviewers' convenience.

- **Reviewer #1 (Remarks to the Author):**

Godwin et al. discovered UBP5, one previously identified PRC2-interacting protein, as histone H2A deubiquitinase. The H2Aub level is increased in *ubp5* mutant. The authors also reported increased H3K27me3 in *ubp5* mutant, and concluded that UBP5-mediated H2A deubiquitination prevents H3K27me3 deposition. Linking H2Aub mediated by PRC1 and H3K27me3 mediated by PRC2 is an interesting issue. However, I have several major concerns about the data analysis and interpretation.

We would like to thank the reviewer for the encouraging comments.

1. The underlying mechanism is unclear. SWN is the catalytic subunit of PRC2, responsible for depositing H3K27me3. While UBP5, which interacts with SWN, prevents H3K27me3 deposition. Whether the prevention of H3K27me3 deposition by UBP5 is independent of SWN interaction?

At this stage, we cannot discard both possibilities: that the prevention of the deposition of H3K27me3 is dependent on or independent of UBP5-SWN interaction. Nevertheless, our results demonstrate that the two proteins directly interact and that UBP5 binding sites to the chromatin are significantly enriched in *cis*-elements related to the recruitment of PRC2. Therefore, we favour a model in which the UBP5-SWN interaction has an active role and is key to regulate the deposition of H3K27me3 in a set of PRC2 target genes. This is supported by the fact that in the absence of UBP5 activity we obtain a significant increase in H3K27me3 levels at specific PRC2 target loci. Our model hence proposes that the direct interaction between UBP5, a transcriptional activator, and SWN, a component of a repressor complex, is key for the dynamic regulation of chromatin similarly at it has been proposed for the interplay between PRC2 and histone demethylases (Lu *et al.*, 2011). This is therefore our current working model, supported by our own evidence and the published literature. However, we agree with Reviewer #1 that, as any model, it will require further validation. To clarify this point and propose possible mechanisms for UBP5 counteracting PRC2 activity, we have added new information to the discussion in lines 346-348, 359-361 and 365-369.

2. There are three major conclusions, i) UBP5 is responsible for H2Aub, which seems apparent from the data (despite only the overlap is presented without statistical test); ii) UBP5 prevents H3K27me3 deposition. This is weakly supported by the data without statistical test. Whether UBP5 overexpression would lead to H3K27me3 increase? Iii) UBP5 counteract with PRC2 repression. This is not supported by the transcriptomic data in Fig. 5H. No clear relationship could be observed between H3K27me3 increase and expression decrease.

We would like to thank the Reviewer for pointing out the lack of statistical analyses. Considering the different points discussed by the Reviewer:

i) To clarify our data regarding the role of UBP5 in controlling H2Aub levels new panels to Figures 2 have been added (Fig. 2C-D; please, see later comments to Reviewer #3 about this new figure). Regarding the lack of statistical analyses, we would like to thank once more the Reviewer for pointing it out. We have now included statistical significance between overlapping gene sets computed using the SuperExactTest package (Wang et al., 2015; Supplementary List 8).

ii) Regarding the UBP5 relationship with H3K27me3:

-We prepared a new Figure 5D showing the significant increase in H3K27me3 levels over UBP5 targets in *ubp5* versus WT plants. We used the Wilcoxon rank sum test for statistical analysis.

-Considering that UBP5 prevents the deposition of H3K27me3 in a set of its target genes, one could expect that overexpression of *UBP5* decreases H3K27me3 in those targets, instead of increasing it as the reviewer has mentioned. Although this is a fair comment, we believe that this is probably a very simplistic scenario, as the result of the overexpression of *UBP5* may strongly depend on other factors, for instance the stoichiometry of the possible UBP5-related complex. Therefore, we took the decision to not overexpress *UBP5* and instead we created our *UBP5pro::UBP5-eGFP;ubp5* line, in which the expression of *UBP5* mimics *UBP5* endogenous levels in the WT background (Supplementary Figure 3A) complementing the *ubp5* mutant phenotypes (Figure 2A-B, Supplementary Figure 4 and 7D). We carried out all our transcriptomic and epigenomic analyses in this line.

-Interestingly, Supplementary Figure 11B shows that, opposite to what we see in UBP5 direct targets, there is a slight but significant decrease in H3K27me3 levels in non-UBP5 targets (Supplementary Fig. 11B, explained in lines 244-245). Furthermore, we also observe a moderate decrease in H2Aub levels in non-UBP5 targets in *ubp5* (Figure 3G, explained in lines 187-188). We think that these indirect changes could act as an epigenomic compensating mechanism in the nucleus of *ubp5* and overall highlight that the absence of UBP5 activity has a profound impact on the Arabidopsis epigenome causing both direct and indirect effects discussed in lines 332-336.

iii) We would like to thank Reviewer #1 for her/his comments about our H3K27me3 results that prompted us to perform a deeper analysis of these data. To clarify this point, we have:

-included a violin cum box plot as new Figure 5H to demonstrate that downregulated genes in *ubp5* show a significant gain of H3K27me3 opposite to upregulated genes, in which no changes in H3K27me3 levels are observed.

-Figure 5I now includes the scatter plot showing the correspondence between H3K27me3 and gene expression changes between Col-0 and *ubp5* plants. Our results also show that gaining H3K27me3 in *ubp5* does not always implies expression changes, as significant increase in H3K27me3 is observed in non-misregulated genes but still less robust than in downregulated genes (Figure 5H). We have also edited the text to explain the relationship between the gain of H3K27me3 and observed transcriptional changes in *ubp5* (lines 253-264).

We do hope that this new detailed analysis will satisfy Reviewer #1's comments and add a new dimension to the manuscript.

3. The major role of PRC1 is catalyzing H2A ubiquitination, and the effect of PRC2 is recognizing H2Aub and catalyzing H3K27me3. Previous reports indicated that H3K27me3 and H2Aub overlap is limited. The identification of UBP5 responsible for removing H2Aub may help explain this phenomenon. One possible mechanism may be that the interaction of UBP5 and SWN is to remove H2Aub and facilitate H3K27me3 at certain loci. Given the genome-wide influence of PcGs, it is difficult to be explained by a single model in Fig. 6. The authors may perform genetic experiment to elucidate the genetic interaction between PRC1 and UBP5, in order to see if UBP5 is antagonistic of the effect of PRC1. Focusing on the interaction between H2Aub and deubiquitination seems more direct and straightforward.

We agree with the reviewer that UBP5 contribution to the independent roles of PRC2 and PRC1 in uncoupling H3K27me3 and H2Aub is a very attractive idea and we have now added lines 346-348 and 354-361 in the discussion commenting this possibility. However, we also observed that in a significant number of UBP5 target genes (640 genes, Figure 5F), the effects of UBP5 upon H2Aub are coupled to changes in H3K27me3 and this is what we highlight in our model. In addition, the Reviewer also proposes another possible mechanism: the interaction of UBP5 and SWN may allow to remove H2Aub and facilitate H3K27me3 at certain loci. In that case, one would expect that in *ubp5* plants, H2Aub gain will be linked to a decrease in H3K27me3 at UBP5 targets. However, our data does not support this hypothesis as it is shown in the following supporting figure:

As the reviewer, we think that a possible functional UBP5-PRC1 interaction is a very exciting idea worthy to be investigated; hence, we would like to thank the reviewer for this comment that supports our own current efforts. Indeed, we have preliminary results indicating that UBP5 may interact with PRC1. However, we do think that clarifying this interaction and its impact on PRC1 activities will require a whole new set of experiments, genetic experiments, as the reviewer points, but also epigenomic, transcriptomic and proteomic experiments that are indeed the focus of one of the current projects in our lab.

4. The title need to be re-considered, which indicates UBP5 is associated with stress responses, but merely based on GO enrichment analysis without experimental validation.

We agree with the reviewer that the title needs rephrasing to better reflect the main results from our study and we have changed it to the following new title: “The UBP5 histone H2A deubiquitinase counteracts PRC2-mediated repression to regulate Arabidopsis development”.

5. Line20, “there is limited knowledge about PRCs’ interacting proteins and their interplay with PRCs in epigenome reshaping, which is fundamental to understand gene regulatory mechanisms”. There are already a large number of PRC interacting proteins reported, as reviewed by authors of this manuscript (10.3390/epigenomes6010008).

We have removed this sentence and added the following one in lines 21-23: “In recent years, there has been increasing evidence of the complexity of PRCs’ interaction networks and their interplay of these interactors with PRCs in epigenome reshaping, which is fundamental to understand gene regulatory mechanisms”.

6. line 121 shows that UBP5pro::UBP5-eGFP can restore the phenotype, and the subsequent analysis lacks the data of UBP5pro::UBP5-eGFP.

We agree with the reviewer that a more detailed analysis of our *UBP5pro::UBP5-eGFP;ubp5* line was missing from our previous manuscript version. Therefore, we have now added:

-Figure 2B, demonstrating that hypocotyl and root length is restored in the *UBP5pro::UBP5-eGFP;ubp5* line when compared to *ubp5*.

-Supplementary Figure 4, showing how expression of specific *ubp5* misregulated genes which are *UBP5* direct targets is complemented to WT levels in the *UBP5pro::UBP5-eGFP;ubp5* line.

-Supplementary Figure 2B showing confocal pictures of *UBP5-eGFP* nuclear signal in Arabidopsis roots.

-Supplementary Figure 7D, showing that overall H2Aub level is restored to WT level in the *UBP5pro::UBP5-eGFP;ubp5* line.

Other comments:

1. To demonstrate the reproducibility of replicates, data from WT and mutants need to be brought together. WT and mutant data can also correlate well if only pairwise comparisons are performed, as transcriptome and epigenomic data are largely unchanged between samples, with quantitative differences occurring at only a few hundred loci.

Thanks to the Reviewer for his/her suggestion. To correct this issue, we have done pairwise comparison for WT and *ubp5* mutant to demonstrate the robustness of our ChIP-seq replicates. Supplementary Figure 13 shows the applied PCA and correlation plots for H2Aub and H3K27me3 for both genotypes.

2. The conclusions of Fig. 4A-B and Page vi, line 192-194 based on proportion instead of statistical analysis and can not answer whether *UBP5* binding gene is significantly enriched in down-regulated expression genes. And the ratio of overlap should not only be compared with up-regulated genes, but also with all genes. The results of Fig4D also lack comparisons with genes without expression changes.

Following reviewer's suggestions, we have now added Wilcoxon rank sum test to Figure 4D and comparison to upregulated, downregulated and all non-misregulated genes in *ubp5*. Our results demonstrate that, in contrast to upregulated genes, downregulated genes show a significant increase in H2Aub. However, an increase in H2Aub does not always results in expression changes as a significant increase in H2Aub is observed in non-misregulated genes but still less robust than in downregulated genes (Figure 4E). This is in line with previously discussed H3K27me3 results. Lines 222-225 in the new manuscript version explain these results. Further, we applied a super exact statistical test to evaluate the significance of all the multi-intersection overlaps (Supplementary Table 8).

3. The conclusion in Page vii, line213-214 "Notably, high H3K27me3 level was particularly pronounced at gene domains corresponding to *UBP5* binding sites " lacks control, i.e., Fig5C-D lacks the situation in non *UBP5* binding sites of the genome as a control, Fig5E, 5G, 5H also lack statistical analysis against the genome-wide background.

We thank once again the reviewer for pointing out the lack of proper statistical analyses for our results. The new version of the manuscript presents the following:

-Figure 5D illustrating the notable rise in H3K27me3 levels over *UBP5* targets when compared to non-*UBP5* targets, serving as the control. The statistical analysis was carried out using the Wilcoxon rank sum test.

-Figure 5F Statistical significance of gene overlap was tested using Super exact test (Supplementary Table 8).

-We have added genes that are non-misregulated to the chart presented in Figure 5H, alongside the upregulated and downregulated genes using the Wilcoxon rank sum test to assess their statistical significance.

-To complement Figure 5E, Supplementary Figure 11C has been added to demonstrate H3K27me3 distribution over non-UBP5 targets as a control.

-For Figure 5I, we consider that there is no apparent need to do additional statistical analysis with genomic wide background in our scatter plot. Moreover, stringent criteria for selection of H3K27me3 and misregulated genes was applied in our bioinformatic pipeline. Further, we have mentioned the correlation coefficient (R) for the plot.

- **Reviewer #2 (Remarks to the Author):**

The manuscript by Godwin et al. described the interaction between UBP5 histone H2A deubiquitinase and PRC2 components and the functions of UBP5 in Arabidopsis. The topic is interesting, but the data is too preliminary to get a conclusion and lack of novelty.

We thank Reviewer #2 for finding the topic of our manuscript interesting. However, we do believe in the novelty of our work as it is presenting pioneering data in the key function of UBP5 as a major H2A deubiquitinase of Arabidopsis. Until now only UBP12/13 proteins had been related with H2A deubiquitination in plants, but the impact of these proteins on the H2Aub epigenetic mark is minor compared to UBP5 as we demonstrate in our manuscript (Supplementary Figure 12). In addition, although in other organisms proteins of the USP/UBP family have previously been shown to regulate H2A deubiquitination, such as USP11 (Ting *et al.*, 2019), to the best of our knowledge our manuscript has been the first one showing epigenomic data that demonstrate the overall impact on H2A deubiquitination by a protein of this family both in plants and animals. Furthermore, our data about the interplay between UBP5 and PRC2-mediated regulation add a new layer to the complexity of PcG regulation. Finally, we hope that the improved version of our manuscript will finally convince Reviewer #2 of the importance of our results for a broad scientific community interested in understanding how developmental and epigenetic regulation are interlocked by chromatin dynamics.

The major concerns:

1, the interaction between UBP5 and PRC2 is not solid.

A, for FRET assay, usually the cutoff for FRET efficiency to determine the interaction between two proteins is 5%. However, the efficiencies for CLF-PWO1 and UBP5-PWO1 is less than 5%;

To the best of our knowledge, we could not find any reference stating that FRET efficiencies below 5% are a standard cutoff for protein-protein interactions analysed through this technique. Our FRET-APB analyses using a LSM780 included all the require negative controls and as positive control we used the CLF-PWO1 interaction, which has previously been published (Mikulski *et al.*, 2019). All interactions checked in our FRET experiments were clearly statistically relevant over the negative controls.

B, for Y2H assay,

1, it is not clear for the condition (medium) for yeast growth in the test group (it shows in the Figure 1D that the medium is WLH lack, but it is indicated that a LWAH lack medium in the Figure legend),

We would like to thank the reviewer for pointing out this mistake. We have now corrected the legend.

2, both UBP5-BD and SWN Δ Set-AD are self-activated, thus it not suitable for Y2H assay;

We agree with the reviewer that there is a very slight growth of our Y2H negative controls on the selective medium, but we hope that the reviewer will agree with us that the tested interactions significantly allow for a much better growth of the yeast clones, indeed clearly surpassing the growth shown by the negative controls.

3, it is not clear why use SET domain-deleted SWN but not full length SWN;

The SWN Δ SET construct that we have used in our Y2H experiments was firstly published in (Chanvivattana *et al.*, 2004) and previously used in our lab (Hohenstatt *et al.*, 2018). However, to validate our results, we tested the interaction of a full SWN construct (kindly shared by Dr Miguel de-Lucas, Durham University) with UBP5 in the same Y2H system:

These new Y2H results confirm that our previous results are identically valid to demonstrate the UBP5-SWN interaction (please, also see answer to Reviewer #3's comment 2).

4, it is strange and unusual why both un-diluted UBP5-BD and un-diluted SWN-AD exhibited strong self-activation, while diluted UBP5-BD and diluted SWN-AD exhibited no self-activation;

We have performed new Y2H experiments to solve this issue (Figure 1D).

C, for Co-IP assay,

1, only detected the interaction between the two proteins under overexpression tobacco cells, not in planta Co-IP assay;

We agree with the reviewer that our co-IP experiments have been performed in a heterologous system in which overexpression of the required constructs was induced in *N. benthamiana*. However, creating the required stable Arabidopsis lines would have required a much longer time and hence we decided to apply this approach that helped us to confirm our Y2H interaction data.

2, It is not sure that whether anti-GFP antibodies can distinguish GFP and mCherry or not;

In our lab we are very familiar with these antibodies as we frequently used them for our protein-protein studies due to the absence of cross-reactivity (Hohenstatt *et al.*, 2018). Please, see our response to the following concern for further clarification.

3, the controls are missed in the Co-IP assay (loading control, express mCherry but not recognized by anti-GFP antibodies in both IP and Co-IP assays). In addition, the authors observed that UBP5 interacts

with PWO1, SWN, CLF, EMF2,VRN2, FIE, MSI1, and BMI1B under their conditions. It is not clear why a predicted deubiquitinase interacts with so many proteins? Therefore, take all these in together, it can not rule out the possibility that the interaction between UBP5-PWO1 (and PRC2) is non-specific.

We would like to apologise to Reviewer #2 because in the previous version of the manuscript the following information was not clear enough. Our protein-protein assays demonstrate an interaction between the following proteins UBP5-PWO1, UBP5-SWN and UBP5-EMF2. While our epigenomic data shows that UBP5 directly targets thousands of genes, among them *CLF*, *EMF2*, *VRN2*, *FIE*, *MSI1* and *BMI1B*. We have now rewrite lines 199-204 to clarify this point.

As commented before, the anti-GFP and anti-mCherry antibodies are routinely used in our group for co-IP analyses due to the lack of cross-reactivity as the Reviewer can see in the following figure:

***N. benthamiana* leaves were co-infiltrated with *i35S::SWNΔSET-GFP* or *i35S::UBP5-mCherry* and after β -estradiol induction, total proteins were extracted and protein samples were analysed by western blot. The membranes were developed with anti-GFP or anti-mCherry to demonstrate the lack of cross-reactivity. Even when the gel was overloaded with protein extract (picture on the right) the respective antibodies did not show any cross-reactivity.**

2, the data in Figure 2 does not support the conclusion that UBP5 is an essential plant stress response regulator. There is no stress phenotype observed in *ubp5* mutant, GO category does not full support the stress response.

We agree with the Reviewer that this is a fair point as our transcriptomic result, although promising, is by itself not sufficient to demonstrate a role of UBP5 in the regulation of stress responses. Therefore, this part has been tuned down (e.g., removed from general title and result chapter 2 title) in the new version of the manuscript and we have focused on UBP5 as a regulator of plant development.

3, there is no direct evidence to support the conclusion that UBP5 acts as a H2A deubiquitinase. The data suggests that UBP5 is required for H2A deubiquitination, but no deubiquitinase activity for UBP5 is provided.

We carried out *in vitro* deubiquitination experiments with bacterial purified UBP5 and commercial histones, however our experiments did not yield the expected results. We think that this is because UBP5 may require a specific environment to carry out its activity. Confirming our hypothesis, when we induced *UBP5* overexpression in *N. benthamiana* epidermal leaf cells, following Reviewer #3's suggestion, this was sufficient to trigger a strong decrease in H2Aub levels (new Figure 3H). Therefore, considering the new evidence and our epigenomic results showing how the *de-novo* H2Aub peaks in *ubp5* perfectly mirror UBP5 binding as well as the published results demonstrating the deubiquitinase

activity of UBP5 (Rao-Naik *et al.*, 2000), our working model proposes that the most probable scenario is that UBP5 directly deubiquitinates H2A (Figure 6).

4, as H3K27me3 is dependent on H2Aub, therefore it is not surprise that UBP5-mediated H2Aub deubiquitination prevents deposition of H3K27me3.

We thank Reviewer 2 for this comment; however, to say that H3K27me3 deposition depends on H2Aub clearly underestimates the complexity of the Polycomb Group (PcG)-mediated regulation that us and other colleagues have discussed in the past (Godwin and Farrona, 2022; Merini and Calonje, 2015; Zhou *et al.*, 2017; Kraleman *et al.*, 2020). As we commented in lines 57-60 of our introduction, “This is also reflected in their activities as H3K27me3 can precede H2Aub (i.e., hierarchical model) or oppositely follows this modification on the chromatin. Furthermore, both marks can independently regulate different set of genes”. Therefore, we think that our data is essential to disentangle the intricate relationship between the removal of both marks, something that has not being well explored in the past, and shows that in a set of genes UBP5-mediated H2Aub prevents deposition of H3K27me3 (discussion in lines 346-348 and 354-361).

- **Reviewer #3 (Remarks to the Author):**

The manuscript by Godwin *et al.* provides evidence of a new PRC2 binding partner that is responsible for deubiquitination of the PRC1 mark H2Aub in Arabidopsis. Phenotypic analysis of *ubp5* mutant demonstrates the role of UBP5 in regulation of Arabidopsis development. RNAseq experiment shows significant transcriptomic alterations in *ubp5* mutant compared to WT plants. In line with other UBP family members, UBP5 is involved in deubiquitination of histone H2A as revealed by western blot and H2AK121ub ChIPseq analysis. The authors propose a model whereby UBP5 protein binds to the 5' end of Arabidopsis PRC target genes to relieve H2A-mediated repression thus promoting gene reactivation. At a number of PRC2 target genes, UBP5 may additionally prevent deposition of H3K27me3. Overall, this novel research delivers a new component taking part in PRC regulation in Arabidopsis. More specifically, the data presented supports the finding of a new eraser of H2Aub likely also affecting deposition of H3K27me3. I have some comments to improve this work.

We thank Reviewer #3 for the supportive comments highlighting the novelty of our research.

Major points

1- Figs. 1A-B show UBP5 nuclear localisation in *N. benthamiana*. To improve this work, the authors could provide evidence of nuclear localisation in *A. thaliana* using the *UBP5pro:UBP5-eGFP* line generated in this work. Given the phenotypes of the *ubp5* mutant reported in Fig. 2A-B, I wonder whether UBP5-GFP expression is restricted to certain tissues of the plant (i.e. meristematic tissues). Fig. 1A could also be improved, it is not clear what the bright field is representing. I would suggest providing a wider view in *N. benthamiana* capturing several cells with UBP5 nuclear localisation.

As proposed by Reviewer #3, we have now added Supplementary Figure 2B showing UBP5-GFP nuclear localisation in Arabidopsis using our *UBP5::UBP5-eGFP;ubp5* line.

Regarding the expression of *UBP5* in WT plants, we have added Supplementary Figure 2A showing expression data for *UBP5* in different plant organs. The results demonstrate that *UBP5* is expressed everywhere in the plant, although the highest expression was observed in siliques. We have now removed the bright field in Figure 1A as we agree with the Reviewer that did not add much to the figure.

2- The authors show interaction of UBP5 with the PRC2 component SWN. However, a mutated version of SWN lacking the SET domain has been used both in Y2H and coIP assays. To further support the interaction of UBP5 and SWN, I would suggest testing UBP5 interaction with full length SWN.

Please, see previous answer to Reviewer #2 regarding this same point and our Y2H data showing interaction between UBP5 and the full SWN. As UBP5-full SWN Y2H new interaction results mimics our previous data, we think that our Y2H results with the SWN Δ Set construct are equally valid. In addition, the SWN Δ Set construct is a published one (Chanvivattana *et al.*, 2004; Hohenstatt *et al.*, 2018) while the unpublished full SWN construct was kindly donated by Dr Migue de-Lucas to confirm our results.

3- Related to the Y2H experiment (Fig. 1D and Suppl fig. 1), UBP5-BD has been used to test interaction with PWO-AD and SWN Δ SET-AD, whereas UBP5-AD has been tested against EMF2-BD. Please provide the reciprocal combinations for all interactions. In addition, please edit suppl. Fig. 1. I understand that the authors have included PWO1-SWN Δ SET and PWO1-UBP5 as positive controls for the EMF2-UBP5 interaction, but actually, these are exactly the same images as in Fig. 1D, not an independent set of positive controls.

We would like to thank the Reviewer for this comment. EMF2-UBP5 and SWN Δ SET-UBP5 experiments with all respective controls were run in parallel, therefore we have now merged the results in the same Figure 1D. Reciprocal combinations for all our Y2H interactions are also shown in Figure 1D.

4- The de-repression of PRC2-regulated genes such as STM, WUS and KNATs (Suppl. Fig. 4) correlates well with the phenotypes observed in the *ubp5* mutant (Fig. 2A). However, given the function proposed for UBP5 as a H2Aub deubiquitinase that prevents PRC2 activity (Fig. 6), de-repression of these developmental genes seem to be an indirect effect of absence of UBP5. I was wondering if the authors would be able to identify genes among the UBP5 targets, whose H2Aub and/or H3K27me3 levels are affected in the *ubp5* mutant, that may better explain the observed phenotypes.

As our previous RT-qPCR analyses had been performed in genes that were mis-regulated in *ubp5* but probably due to indirect effects, to perform a more meaningful analysis, we have added a new Supplementary Figure 4 to this new version of the manuscript showing RT-qPCR data for the changes in expression of specific UBP5 target genes which gained H3K27me3 and H2Aub in the *ubp5* mutant. The selected misregulated UBP5 target genes may contribute to explain some of the phenotypes observed in *ubp5* plants as *SAMBA* plays a major role in pollen development and organ size control, *UPP* is involved in maintenance of root morphology, *GAF1* participates in female gametophyte development and male competence, *ACT1* plays a major role in pollen tube development and *GC4* is involved in maintaining Golgi structure and has a possible role in plant growth and development.

5- This study proposes a direct role of UBP5 in H2Aub deubiquitination of numerous targets (Fig. 3B-H) thus having an effect on global levels of H2Aub and H3K27me3 (Fig. 3A and 3B). However, one could not rule out an indirect effect of UBP5 on the mentioned histone modifications. Since the authors count with an inducible i35S:UBP5-GFP construct, it would be interesting to test whether inducing UBP5 expression would have a global effect on H2Aub levels in the *N. benthamiana* system.

Thank you very much for proposing this approach. As suggested by the Reviewer, these experiments were performed, and results are now shown in the new Figure 3H. The results demonstrate that the inducible overexpression of *UBP5* in *N. benthamiana* epidermal leaf cells is sufficient to induce a strong decrease in overall H2Aub levels.

6- Regarding Fig. 3B, the description of the different categories presented in the Venn diagram could be improved. It would also be helpful to point here which are the “hyper-marked” and the “de novo-marked” genes. It is not clear to this reviewer what is the difference between the 1,818 and the 4,383 genes under the *ubp5* diagram.

We agree with Reviewer #3 that that the original Figure 3B was not clear enough to summarise our epigenomic results. Therefore, we have now added figures 3B, C and D. Figure 3B compares H2Aub levels between WT and *ubp5* plants by MACS3 peak calling highlighting that in total there are 21,017 genes showing a significant H2Aub peak in *ubp5*. Among these genes, 6,201 are exclusively marked by H2Aub in *ubp5* (i.e., *de-novo* H2Aub); whereas 14,816 genes were marked by H2Aub in both WT and *ubp5* mutant plants. Figure 3C shows that, among these 14,816 H2Aub marked genes, 3,055 genes showed a significant increase in H2Aub signal in *ubp5* (i.e., hyper-marked in *ubp5*), 4,611 did not show any changes between WT and *ubp5* plants (i.e., unchanged) and 7,150 showed a significant decrease in H2Aub signal in *ubp5* (i.e., hypo-marked in *ubp5*). Therefore, these results strongly suggest that UBP5 has a paramount role in controlling H2Aub nuclear dynamics. Finally, Figure 3D shows that among all these epigenetic changes, there is a significant overlap between UBP5 target genes with the *de-novo* and hyper-marked genes (i.e., gained in H2Aub).

7- The discussion section is quite long, covering aspects of the UBP5 orthologs in plants and animals as well as the possible role of UBP5 in stress responses and in the PRC1/PRC2 dynamics. These are all indeed very exciting topics and it's worth mentioning them. Yet, I would suggest trying to focus more on linking these aspects to the phenotypes observed in the *ubp5* mutant.

Following Reviewer #3's suggestions, we have now shortened and have re-focused the discussion.

Minor points

- Lines 74-77: The enrichment of GAGA and Telobox motifs over UBP5-GFP binding sites is indeed interesting and provides a good link to PRC2 regulation. However, referring to UBP5 as a “sequence-specific eraser” may not be appropriate. It sounds like UBP5 is able to bind DNA by itself.

We fully agree with the reviewer and have rephrased this sentence in lines 83-85 to “The vast majority of UBP5 direct target genes showed either hyper-marking or *de-novo* marking by H2Aub in *ubp5* plants, altogether indicating that UBP5 acts as an eraser of this epigenetic mark”.

- Line 112, “UBP5 is an essential plant developmental and stress response regulator”. Considering the data presented in this work, I agree that UBP5 is required for Arabidopsis development, but I am not convinced about its role in stress responses. The authors are not providing experiments to support UBP5 function in stress responses.

As per our previous answer to Reviewer #2, this part has been tuned down in the new version of the manuscript and we have focused on UBP5 as a regulator of plant development.

- According to the text (lines 106-107), SWN Δ SET was used in colP experiments. Please correct SWN-GFP in Fig. 1E.

Corrected.

- Line 573, Fig. 1 legend: please correct FRET-APB

Corrected.

- Legend of Suppl. Fig. 4: remove ns as it does not appear in any of the plots.

Suppl. Fig. 4 has been removed from this new version and substituted by the new Suppl. Fig. 11.

References:

- Wang, M., Zhao, Y. & Zhang, B.** Efficient test and visualization of multi-set intersections. *Sci. Rep.* 5, 16923 (2015). **Chanvivattana Y, Bishopp A, Schubert D, Stock C, Moon YH, Sung ZR, Goodrich J 2004.** Interaction of Polycomb-group proteins controlling flowering in Arabidopsis. *Development.* 5263-5276.
- Hohenstatt ML, Mikulski P, Komarynets O, Klose C, Kycia I, Jeltsch A, Farrona S, Schubert D 2018.** PWWP-DOMAIN INTERACTOR OF POLYCOMBS1 interacts with polycomb-group proteins and histones and regulates arabidopsis flowering and development. *Plant Cell.* 117-133.
- Lu F, Cui X, Zhang S, Jenuwein T, Cao X. 2011.** Arabidopsis REF6 is a histone H3 lysine 27 demethylase. *Nat Genet* 43(7): 715-719.
- Mikulski P, Hohenstatt ML, Farrona S, Smaczniak C, Stahl Y, Kalyanikrishna, Kaufmann K, Angenent G, Schubert D 2019.** The chromatin-associated protein pwo1 interacts with plant nuclear lamin-like components to regulate nuclear size. *Plant Cell.* 1141-1154.
- Rao-Naik C, Chandler JS, McArdle B, Callis J. 2000.** Ubiquitin-specific proteases from Arabidopsis thaliana: cloning of AtUBP5 and analysis of substrate specificity of AtUBP3, AtUBP4, and AtUBP5 using Escherichia coli in vivo and in vitro assays. *Arch Biochem Biophys* 379(2): 198-208.
- Ting X, Xia L, Yang J, He L, Si W, Shang Y, Sun L. 2019.** USP11 acts as a histone deubiquitinase functioning in chromatin reorganization during DNA repair. *Nucleic Acids Res* 47(18): 9721-9740.

REVIEWERS' COMMENTS

Reviewer #1 (Remarks to the Author):

The authors addressed my major concerns. I have no further comments.

Reviewer #2 (Remarks to the Author):

The revised manuscript is much improved, but still not fully addressed my major concerns for the physical interaction between UBP5 and PRC2 (SWN), and whether UBP5 itself is a H2A deubiquitinase.

1, The physical interaction between UBP5 and PRC2 (SWN) is the foundation for this work. Current data suggests that their physical interaction from both FERT and Y2H assays are weak, and the Co-IP is performed in a heterologous system in *N. benthamiana* with overexpression of the proteins. To validate their physical interaction in vivo, the co-IP assay in *Arabidopsis* is required.

2, As mentioned by the authors, the novelty of this work is that it presents pioneering data in the key function of UBP5 as a major H2A deubiquitinase of *Arabidopsis*. However, no direct evidence is provided to support UBP5 as a H2A deubiquitinase, the current evidence suggest that UBP5 is required for H2A deubiquitination. Therefore, the assay for H2A deubiquitinase activity for UBP5 is required.

Reviewer #3 (Remarks to the Author):

I appreciate that the authors have answered all the points that I raised. I don't have any further comment.

- **Answer to reviewers' comments:**

We are pleased that all concerns raised by Reviewers #1 and #3 have now been addressed. Regarding Reviewer #2's last comments:

- The revised manuscript is much improved, but still not fully addressed my major concerns for the physical interaction between UBP5 and PRC2 (SWN), and whether UBP5 itself is a H2A deubiquitinase.

Many thanks to the Reviewer for recognising our efforts to improve the manuscript. However, we think that our data provide strong evidence of the role of UBP5 as a H2A deubiquitinase counteracting PRCs' activities. See below further comments to support this statement.

- The physical interaction between UBP5 and PRC2 (SWN) is the foundation for this work. Current data suggests that their physical interaction from both FRET and Y2H assays are weak, and the Co-IP is performed in a heterologous system in *N. benthamiana* with overexpression of the proteins. To validate their physical interaction in vivo, the co-IP assay in *Arabidopsis* is required.

As previously indicated, the UBP5-PRC2 interaction has now been tuned down and the use of heterologous systems have been commented in the new version of the manuscript.

- As mentioned by the authors, the novelty of this work is that it presents pioneering data in the key function of UBP5 as a major H2A deubiquitinase of *Arabidopsis*. However, no direct evidence is provided to support UBP5 as a H2A deubiquitinase, the current evidence suggest that UBP5 is required for H2A deubiquitination. Therefore, the assay for H2A deubiquitinase activity for UBP5 is required.

We consider that our combined data provides sufficient evidence of the role of UBP5 as a major H2A deubiquitinase in *Arabidopsis*. In addition, we also discuss recent works by Zheng et al. confirming our results.